# Formulation of silages from spent mushroom substrates of *Pleurotus ostreatus* and *Lentinula edodes*: Organoleptic properties, phenolic content, *in vitro* digestibility, gas production and ruminal kinetics

Angélica Valeria Lorenzana-Moreno[1☉], Diana Victoria Valdés-Meléndez[2☉],
Hermilo Leal Lara[3], José Moisés Talamantes-Gómez[2],
Augusto César Lizarazo-Chaparro[1], Claudia C. Márquez-Mota[2]*

1 Centro de Enseñanza Práctica e Investigación en Producción y Salud Animal, Facultad de Medicina Veterinaria y Zootecnia, UNAM, Ciudad de México, México, 2 Departamento de Nutrición Animal y Bioquímica, Facultad de Medicina Veterinaria y Zootecnia, Universidad Nacional Autónoma de México, Ciudad de México, México, 3 Departamento de Alimentos y Biotecnología, Facultad de Química, Universidad Nacional Autónoma de México, Ciudad de México, México

☉ These authors contributed equally to this work.
* c.marquez@unam.mx

## Abstract

To fulfill the global demand for sustainable livestock production and the implementation of circular economy models, the search for alternative feed sources to lower production cost has increased significantly. The use of agro-industrial waste has proven to be a low-cost strategy for animal feed. The present study evaluates the use of spent mushroom substrate (SMS) from *Pleurotus ostreatus* (strain Po-IAP) and *Lentinula edodes* (strain L5) as an ingredient for silage. A total of eight micro-silages were formulated using SMS and ground yellow corn in the following proportions (%SMS: % corn): 100:0, 90:10, 80:20 and 70:30. Differences in the nutritional composition, secondary metabolite content, *in vitro* digestibility (IVDMD), and fermentation parameters were evaluated. Organoleptic analysis showed that silages with 70% SMS had better color and odor profile compared with silages with 80, 90 and 100% SMS. *Lentinula edodes* silages had the highest content of phenolic compounds (8.2–9.0 mg GAE/g DM) compared with *Pleurotus ostreatus* (strain Po-IAP) silages. Silages with 70% SMS inclusion had higher IVDMD. Silages with 70% *Lentinula edodes* (strain L5) SMS had the highest gas production and Silages with 70% *Pleurotus ostreatus* (strain Po-IAP) SMS showed a shorter lag phase. Overall, the results obtained in the present study indicate that the formulation of silages with 70% of SMS had good organoleptic characteristics and nutritional qualities that improves IVDMD, and fermentative parameters and they therefore could be used as animal feed. Further, *in vivo* studies are recommended to fully evaluate the possible health effects of these silages on animal health and to evaluate its impact on production cost.

**Data availability statement:** All data are in the paper and supporting information files.

**Funding:** This research was financially supported by grant: PAPIIT IN212822, from the Dirección General de Asuntos del Personal Académico (DGAPA)- UNAM. The funders had no role in study design, data collection and analysis, decision to publish, or preparation of the manuscript.

**Competing interests:** The authors have declared that no competing interests exist.

## Introduction

Recently, global interest in economic development and circular economy has increased significantly to enhance sustainable food production and manage agro-industrial waste more effectively. Circular economy models focus on reducing waste generation and enhancing the use of resources by repurposing by-products into valuable inputs for other systems [1]. For instance, in animal feed. By-products such as citrus pulp, grape pomace, olive leaves, mango and avocado residues have been successfully used in the animal feed of pigs, poultry and ruminants [2–4].

The 2030 Agenda for Sustainable Development states that actions must be taken to battle climate change, promote health and education, reduce poverty, and boost economic growth [5,6]. One strategy used to increase economic growth following circular economy principles is the production of edible mushrooms. Edible mushrooms like *Pleurotus ostreatus* and *Lentinula edodes* can grow on several agro-industrial waste, such as corn stover [7], crop residues, cotton stalks [8], paper scraps and spent coffee grounds [9], etc. These characteristics make mushroom cultivation not only environmentally sustainable but also economically feasible.

It is predicted that the global market for mushrooms will grow exponentially over the next five years [10]. However, the increase in mushroom production will lead to a rise in the generation of by-products, specifically spent mushroom substrate (SMS) [11]. It is estimated that for every kilogram of produce mushroom, a total of 5 kg of SMS is generated [12]. This by-product is a mixture of modified lignin and mycelium residues, rich in protein, cellulose, chitin, and bioactive compounds [13]. SMS has been successfully used as fertilizer, for substrate in the cultivation of other edible mushrooms, as a biosorbent, and, due to its nutritional properties, as animal feed [14–16]. The use of SMS in livestock feeding may reduce production expenses and decrease the use of conventional feed ingredients.

Several studies have demonstrated the SMS could be used as a feed source for livestock [12]. Previous *in vitro* studies in sheep demonstrated that SMS from *Pleurotus ostreatus* and *Lentinula edodes* significantly enhances the *in vitro* dry matter digestibility (IVDMD) up to 47.7% [7]. However, to efficiently use SMS as animal feed, it is necessary to preserve it. SMS is a residue with a 60% moisture content; therefore, ensiling the material might enhance its shelf life and preserve its nutritional quality. Fermentation of SMS with cellulolytic bacteria such as *Enterobacter* spp. and *Bacillus* spp improved growth performance and carcass trait of Hanwoo steers [14], indicating that the fermentation of the SMS does not affect its nutritional value and that it could be used as a replacement for conventional roughage.

It has been well documented that fruiting bodies of fungi are an important source of secondary metabolites (SM) with several biological activities such as antioxidant, anti-cancer, anti- inflammatory among others [17]. However, less studies have been published bout the presence of SM in SMS, recent studies of these compounds in SMS are mainly focused on its antibacterial, antifungal and antiparasitic activity [18,19] yet their potential effects on digestibility and fermentative parameters in ruminants should be further explored.

The benefits of fungal treatment of lignocellulosic biomass have been documented in recent studies. For instance, Vorlaphim et al [20] demonstrated that rice stubble treated with *Pleurotus ostreatus* and urea enhances the nutrient digestibility and growth in slow-growing goats. Similarly, a study performed on Awassi Lambs showed that the SMS could replace up to 15% of barley without changes in carcass characteristics [21]. In a study conducted in Berari Goats it was observed that the inclusion of 10% of SMS in a concentrated mixture improved body condition score, blood glucose, and hemoglobin levels indicating its potential to enhance animal growth performance [22]. Even though the effectiveness of SMS on animal nutrition has been well documented, less studies have focused on the silage of dried residues. As previously stated, due to its high moisture content, ensiling SMS is recommended to improve preservation and feeding safety. It has been demonstrated that ensiling high moisture content biomass enhances fermentation quality and nutrient retention [23].

We hypothesized that preservation of SMS through ensiling will increase its nutritional quality and would not affect the SM content, improving its digestibility. In the present study the SMS from *Pleurotus ostreatus (*strain Po-IAP) and *Lentinula edodes* (strain L5) were used due to its capacity to grow on substrates formulated with corn stover, as well, for their nutritional value, and documented bioactive properties, making their residues particularly suitable for valorization in animal nutrition. Thus, in the present study we evaluated the nutritional quality, SM content, *in vitro* digestibility and fermentative parameters of different SMS silages from *Pleurotus ostreatus* (strain Po-IAP) and *Lentinula edodes* (strain L5).

## Materials and methods

### Animal management and treatments

The research protocol (711) used in the present study was reviewed and approved by the Internal Committee for the Care and Use of Animals (CICUA) of the Faculty of Veterinary Medicine and Zootechnics, National Autonomous University of Mexico (FMVZ, UNAM). Experiments were carried out at the Center for Practical Teaching and Research in Animal Production and Health (CEPIPSA).

### Collection of spent mushroom substrate (SMS) and silage formulation

For the preparation of silages, residual substrates from the production of the mushrooms *Lentinula edodes* (strain L5) and *Pleurotus ostreatus (*strain Po-IAP*)* were used. The cultivation substrate comprised 80% corn stover, 6% ground sorghum, 4% wheat bran, 5% corn gluten, 4% $CaCO_3$ and 1% $CaSO_4$, which were prepared, sterilized and inoculated in the facilities of a commercial company (Hongos Leben, State of Mexico, Mexico). Once inoculated, the substrates were taken to the Department of Food Science and Biotechnology, Faculty of Chemistry (UNAM) and placed in the fruiting room as previously reported [7].

After fructification, the SMS were taken to the Center for Practical Teaching and Research in Animal Production and Health (CEPIPSA, FMVZ-UNAM) where a total of eight silages were formulated (Table 1). The silages were mixed on a smooth, clean concrete surface measuring approximately 2 x 2 meters, using a shovel. The mixture was based on SMS, to which the appropriate amount of ground yellow corn was gradually added. Once a completely homogeneous mixture was achieved, it was placed into white plastic containers with an approximate capacity of 2.2 kg per silage. The containers were filled in layers of approximately 5 cm in thickness, with each layer being compacted before adding the next. Once the container was full, the maximum amount of air was removed using a vacuum pump, and the lid was sealed with hot silicone.

The inclusion of the ingredients was established on a wet basis, with a total of 32 micro- silages of approximately 3 kg where elaborated (4 replicates for each treatment). All micro-silages were kept in the shade, in a dry place, and at room temperature until they were opened after 35 days.

**Table 1. Silage formulation.**

| Treatment | SMS | | Ground yellow corn |
|---|---|---|---|
| | *Lentinula edodes* (strain L5) | *Pleurotus ostreatus* (strain Po-IAP) | |
| | Composition (%) | | |
| L100 | 100 | | 0 |
| L90 | 90 | | 10 |
| L80 | 80 | | 20 |
| L70 | 70 | | 30 |
| P100 | | 100 | 0 |
| P90 | | 90 | 10 |
| P80 | | 80 | 20 |
| P70 | | 70 | 30 |

## Analysis of the micro-silages

**Organoleptic characteristics and pH of the micro-silages.** The organoleptic characteristics and pH of the micro-silages were evaluated immediately upon opening. The organoleptic characteristics of micro-silages were evaluated through a qualitative/ quantitative criterion. For such purposes, a score system is proposed based on the presence or absence of certain characteristics of the sample. Six variables were evaluated: moisture (M), fungal spots (FS), color (C), particle size (PS), odor (O) and spots between layers (SBL) each variable was scored as 0.5 (inadequate), 0.75 (regular) and 1.0 (adequate). According to the criteria established in Table 2.

For pH measurement, 10g of silage mixed with 90 mL of deionized water and incubated 30 min at room temperature, after which the sample was filtered in a 1 mm mesh and pH was measured with OAKTON® potentiometer [24].

**Chemical analysis of micro-silages.** The chemical composition of the micro-silages was analyzed according to the protocols from the AOAC [25]. Neutral detergent fiber (NDF), acid detergent fiber (ADF), lignin, cellulose, and hemicellulose contents were also analyzed [26].

**Determination of mycotoxin by thin layer chromatography (TLC).** Mycotoxin determination was performed according to the protocol from Hyde, W et al., 1977, with slight modifications [27]. Briefly, for mycotoxin extraction, 25g of silage samples were homogenized during 5 minutes with 100 mL extraction buffer of acetonitrile and potassium chloride (9:1 ratio) (acetonitrile: Sigma-Aldrich, cat-34988; potassium chloride: Sigma-Aldrich, cat-P80911). After homogenization the samples were filtered using separatory funnel. To recover the aqueous phase (extract), two consecutive washes with 50mL of petroleum ether (Sigma-Aldrich, cat-673838) were performed.

A decolorizing gel was prepared using 100 mL of deionized water, 10 mL of iron chloride (10%) (Sigma-Aldrich, cat-157740) adjusted to pH 4.2–4.6 with sodium hydroxide (4%) (Sigma-Aldrich, cat-221465). The complete aqueous phase

**Table 2. Organoleptic analysis of silage.**

| Variable | Inadequate (0.5) | Regular (0.75) | Adequate (1.0) |
|---|---|---|---|
| Fungal Spots (FS) | Thick layer on the entire surface | Light layer in certain parts | Absence |
| Color (C) | Black or dark brown | Tobacco brown | Light brown |
| Particle Size (PS) | Greater than 5 cm or less than 1 cm | Greater than 2.5 cm or less than 5 cm | Between 1–2.5 cm |
| Odor (O) | Acidic and irritating | Musty smell | Slightly acidic fermentation, sweet and fermented |
| Spots between layers (SBL) | More than 30% | Less than 20% | Absence |

and decolorizing gel were mixed in a beaker and placed in an orbital shaker (Heathrow Scientific-Digital Orbital Shaker) for 3 minutes. The mixture was then filtered through filter paper.

A final purification was carried out using separatory funnel by adding 50 mL of chloroform (Sigma-Aldrich, cat-319988) to the decolorized sample. After mixing, the chloroform phase was discarded. This step was repeated twice. The residual water was removed by adding sodium sulfate (Sigma-Aldrich, cat-238597). The remaining chloroform was evaporated in a water bath at 36°C while applying nitrogen at a fixed and constant pressure. Finally, the dry samples were resuspended in 500 µL of an acetonitrile and benzene mixture (2:98%) (acetonitrile: Sigma-Aldrich, cat-34988; benzene: Sigma-Aldrich, cat-319953).

For the TLC analysis, a silica plate (Macherey-Nagel ®) was spotted with 30 µL of each extract and standard solutions for Aflatoxin B1 (Sigma-Aldrich, cat-A6636), Zearalenone (Sigma-Aldrich, cat-Z2125), and Ochratoxin A (Sigma-Aldrich, cat-CRM46912). The presence of mycotoxins was observed by UV light after TLC development.

## Quantification of total phenolic compounds tannins of micro-silages

**Sample preparation.** 200 mg of defatted sample was suspended in 10 mL acetone (70%) (J.T. Baker ®), the suspension was sonicated (CS-UB100) for 20 min at room temperature. After sonication, the sample was centrifuged (Thermo Scientific – Sorval Biofuge PrimoR) at 3000 g for 10 min at 4°C. The supernatant was collected in an amber bottle, while the pellet was resuspended in 10 mL of 70% ethanol, and the extraction process was repeated. The supernatant of both extractions was combined in the same amber bottle and stored at 4°C for further analysis [28].

**Measurement of total phenolic compounds.** Total phenolic compounds were determined by spectrophotometry using a modified Folin-Ciocalteu assay [29]. Briefly, 150 µL of extract was mixed with 150 µL of 1:1 Folin-Ciocalteu reagent (Sigma-Aldrich, cat 47641) and incubated in the dark for 5 minutes. After incubation 1.2 mL of sodium bicarbonate (Sigma-Aldrich, cat S8875) was added, samples were mixed and incubated for 45 min in the dark. The quantification was performed using a standard curve of gallic acid (Sigma-Aldrich, cat 398225) at concentrations of 6.25, 12.5, 25, 50 and 100 ug/mL. Absorbance was measured at 760nm using a spectrophotometer (Thermo Scientific G10 UV-vis Bio) at 760 nm. Phenolic content is expressed as a mg gallic acid equivalent per gram of dry matter (mg GAE/gDM).

**Measurement of total tannins.** For tannin measurement, 1 mL extract was mixed with 100 mg of polyvinylpolypyrrolidone (PVPP) (Sigma-Aldrich, cat-77627) and incubated for 15 min at 4°C. After incubation, the samples were centrifuged (Thermo Scientific – Sorval Biofuge PrimoR) at 3000 g for 15 min at 4°C and the supernatant was collected and analyzed as previously described by the Folin-Ciocalteu assay. The quantification was performed using a standard curve of tannic acid (Sigma-Aldrich, cat-403040) at concentrations of 6.25, 12.5, 25, 50 and 100 ug/mL. Absorbance was measured at 760nm using a spectrophotometer (Thermo Scientific G10 UV-vis Bio) at 760 nm. Tannin content is expressed as a mg tannic acid equivalent per gram of dry matter (mg TAE/gDM) [28].

## *In vitro* dry matter digestibility (IVDMD), gas production and fermentation profile of SMS micro-silages

**Sample preparation.** Samples from each micro-silages were dried (50°C) and milled to pass through a 1 mm sieve and stored until analysis.

***In vitro* dry matter digestibility procedure.** The IVDMD was performed according to the technique proposed by Theodorou et al. (1994) [30] with slight modifications by Molho et al., (2022). Briefly, ruminal fluid was collected from five French Alpine goats (50–55 kg, 4 years old), previously conditioned for one week to a diet of SMS silage [31]. The sample was obtained using an esophageal [32] tube after a 12 hour fast, filtered through eight layers of cheesecloth, and then stored under anaerobic conditions at 39°C. To obtain rumen inoculum, rumen fluid was mixed with reduced and mineral solutions [33] in a ratio of 1:9 v/v. For incubation, 500 mg of dry sample was added to polyester bags with 67 µM pore size [34] and placed in 125 mL amber glass bottles (experimental unit), subsequently 60 mL of rumen inoculum was added under $CO_2$ to maintain anaerobic conditions. A total of four bottles per treatment were used. Bottles with rumen inoculum

and blank controls were hermetically sealed with rubber stoppers and aluminum caps and then placed in a water bath with lateral oscillation (30/min) at 39°C. Gas pressure (kg/cm2) was recorded at 0, 3, 6, 9, 12, 18, 24, 30, 36, 42, 48, 60, 72 hours using a digital manometer (Traceable®, Fisher Scientific, USA) [31].

At the end of the incubation stage, the bottles were opened for pH measurement (pH Tester model 30 Double Function®). Subsequently, 8 mL of the medium was taken from each flask and mixed with 2 mL of 25% (w/v) metaphosphoric acid (Sigma-Aldrich, cat-239275). The samples were stored at −20°C until analysis for volatile fatty acids (VFAs).

For IVDMD calculation, the polyester bags were washed with distilled water, dried at 55°C for 24 h, and weighed. IVDMD was calculated according to Equation 1.

$$IVDMD\ (\%) = \frac{(DM_{initial}) - (DM_{residual})}{DM_{initial}} \times 100$$

(1)

**Ruminal kinetics.** The gas pressure readings (kg/cm$^2$) were transformed into volume with the linear regression equation) proposed by Ørskov and McDonald (1979), shown in Equation 2 [35].

$$GasVol = \frac{Pressure}{0.109}; \ R^2 = 0.988$$

(2)

The variables of the gas production kinetics: maximum volume of gas produced (Vmax), lag phase (L) and gas production rate (S), were obtained by means of a logistic model [36].

**Determination of ammoniacal nitrogen and volatile fatty acid (VFA).** Samples preserved in metaphosphoric acid were thawed and centrifuged (Thermo Scientific – Sorval Biofuge PrimoR) at 3000 g for 25 min at 4°C, then decanted into amber glass bottles and kept refrigerated at 4°C until analysis. Ammoniacal nitrogen was determined using the modified McCollough technique (1967) [37], each sample was read at 630 nm (Thermo Scientific G10 UV-vis Bio) and analyzed in triplicate.

Volatile fatty acids were determined by gas chromatography using a Perkin Elmer AutoSystem XL with autoinjector, analyzing each sample in triplicate as previously reported [31].

## Statistical analysis

The data from the organoleptic analysis were evaluated using the Kruskal-Wallis test to assess differences among treatments, revealing a significant effect (p ≤ 0.05). Dunn's test with Bonferroni correction was applied for pairwise comparisons, identifying significant differences (p ≤ 0.05) between treatments. Statistical analyses were conducted using the *dunn. test* function in R (version 4.3.2).

The data from the chemical analysis, total phenolic compounds and total tannins content were analyzed with the agricolae package in R (version 4.3.2). Values were compared by a one-way ANOVA with Tukey's post-hoc test (p < 0.05), according to the following model:

$$Y_{ij} = \mu_i + \iota_i + \varepsilon_{ij}$$

Where $Y_{ij}$ = response variable, $\mu$ = general mean, $\iota_i$ = effect of SMS silage at level i, j, and $\varepsilon_{ij}$ = effect of random error.

The data from the *in vitro* dry matter digestibility, gas production and fermentation profile of SMS micro-silage were analyzed as a 4 x 2 factorial experimental design in a completely randomized design. A two-way ANOVA was performed using the MIXED procedure of SAS (version), the comparison of means was performed using the Tukey test [38]. According to the following model:

$$Y_{ij}k = \mu + A_i + B_j + (AB)_{ij} + E_{ij}k$$

Where: $Y_{ij}k$ = response variable in repetition k, i = 1,2, …a Level J of b, level i of A. $_j$ = 1,2, …b, μ = Overall mean, $A_i$ = Effect of factor A at level I, $B_j$ = Effect of factor B at level j, $(AB)_{ij}$ = Effect of the interaction AB at level i,j and Eijk = Random error

## Results

### Organoleptic characteristics and pH of the micro-silages

ThConclusionse physical and organoleptic characteristics of the silage, such as color, odor, and the presence or absence of fungi, are key indicators for determining its physical quality, which is closely related to the fermentation process. All the formulations used in this study were effective for ensiling SMS, however they showed different organoleptic characteristics (Table 3), all the treatment showed a light layer fungal spots in certain parts, the best color ($p < 0.05$) was observed in the *Pleurotus ostreatus* (strain Po-IAP) micro-silages (P100-P70), there was no difference in particle size among the micro-silages, the micro-silages formulated with *Lentinula edodes* (strain L5) showed the most acidic ($p < 0.05$) and irritating odor of all the treatments. Treatments L90, L80, L70, P80 and P70 did not show spots between layers, meanwhile L100, P100 and P80 showed less than 20% of spots between layers. The micro-silages with the highest score were P80 and P70. pH is also one of the most important indicators of silage quality, as a low pH reflects efficient fermentation and is associated with the inhibition of undesirable microorganisms. The highest pH was observed in L100 and P100 treatments and the lowest pH was observed in L70 and P70 treatments.

The chemical composition of micro-silages formulated with different levels of SMS from *Lentinula edodes (strain L5)* and *Pleurotus ostreatus* (strain Po-IAP) is shown in Table 4. The highest ash content ($P = 0.001$) was observed in P100 (22.1%) and the lowest ($P = 0.001$) in L70 (9.6%). Crude protein content is in a range of 8.3% (L70) to 12.0% (P100). In the case of fiber fractions, the highest ($P = 0.001$) NDF level was observed in L80 (53.7%), and lowest level ($P = 0.001$) was observed in P70 (32.1%). ADF was 2.4-fold higher ($P = 0.001$) in L100 in comparison with L70, which had the lowest ($P = 0.001$) ADF content. Silage with 100% SMS (L100 and P100) had the highest cellulose content ($P = 0.001$) (21.4 and 18.9%, respectively) in comparison with silage with the lower ($P = 0.001$) content of SMS (L70 and P70). The highest ($P = 0.001$) hemicellulose content was observed in L80 (30.7%). The highest lignin ($P = 0.001$) content was observed in L100 and P100 (13.3 and 13.5%, respectively) and the lowest ($P = 0.001$) level in P70 (7.9%).

**Table 3. Organoleptic characteristics of micro-silages from SMS of *Lentinula edodes* (strain L5) and *Pleurotus ostreatus* (strain Po-IAP).**

| Treatment | Organoleptic characteristics | | | | | pH |
|---|---|---|---|---|---|---|
| | FS | C | PS | O | SBL | |
| L100 | 0.75 | 1[a] | 0.88 | 0.5[d] | 0.88[ab] | 5.1[b] |
| L90 | 0.88 | 0.94[a] | 0.88 | 0.5[d] | 1[a] | 4.1[c] |
| L80 | 0.94 | 0.75[b] | 1 | 0.5[d] | 1[a] | 4.0[d] |
| L70 | 0.88 | 0.94[a] | 1 | 0.5[d] | 1[a] | 3.9[e] |
| P100 | 0.75 | 1[a] | 1 | 0.75[c] | 0.75[b] | 5.3[a] |
| P90 | 0.88 | 1[a] | 1 | 0.75[c] | 0.88[ab] | 4.1[c] |
| P80 | 0.81 | 1[a] | 1 | 0.88[b] | 1[a] | 4.0[d] |
| P70 | 0.75 | 1[a] | 1 | 1[a] | 1[a] | 3.9[e] |
| P-value | 0.115 | 0.001 | 0.0715 | 0.001 | 0.001 | 0.001 |
| SEM | 0.02 | 0.017 | 0.014 | 0.034 | 0.019 | 0.093 |

L100–L70: *Lentinula edodes* (strain L5), with 100%–70% mushroom and 0%–30% ground yellow corn. P100–P70: *Pleurotus ostreatus* (strain Po-IAP), with 100%–70% mushroom and 0%–30% ground yellow corn. FS: fungal spots; C: color; PS: Particle size; O: odor; SBL: spots between layers. The values are the mean, n = 4. SEM: standard error of the mean. Means in a column without a common letter differ from each other, p < 0.05. Chemical analysis of micro-silages.

**Table 4. Chemical composition of micro-silages from SMS of *Lentinula edodes* (strain L5) and *Pleurotus ostreatus* (strain Po-IAP).**

| Treatment | Composition (g/100g DM) | | | | | | |
|---|---|---|---|---|---|---|---|
| | Ash | CP | NDF | ADF | Cel | H | L |
| L100 | 19.2[b] | 11.6[ab] | 50.4[ab] | 41.5[a] | 21.4[a] | 9.0[d] | 13.3[a] |
| L90 | 16.5[c] | 11.4[b] | 53.1[a] | 39.6[a] | 16.1[c] | 13.5 cd | 9.3[bc] |
| L80 | 13.4[d] | 9.8[c] | 53.7[a] | 22.9 cd | 12.7[d] | 30.7[a] | 7.1[c] |
| L70 | 9.6[f] | 7.8[e] | 43.1[b] | 17.14 | 7.6[e] | 25.9[ab] | 7.0[c] |
| P100 | 22.1[a] | 12.0[a] | 44.6[ab] | 37.3[a] | 18.9[b] | 7.2[d] | 13.5[a] |
| P90 | 16.6[c] | 11.1[b] | 43.1[b] | 32.1[b] | 17.0[c] | 11.0[d] | 11.6[ab] |
| P80 | 13.1[d] | 10.1[c] | 47.7[ab] | 27.7[bc] | 15.8[c] | 20.0[bc] | 11.5[ab] |
| P70 | 10.6[e] | 9.1[d] | 32.1[c] | 21.4[de] | 11.6[d] | 10.8[d] | 7.9c |
| P-value | 0.001 | 0.001 | 0.001 | 0.001 | 0.001 | 0.001 | 0.001 |
| SEM | 0.72 | 0.25 | 1.54 | 1.61 | 0.76 | 1.72 | 0.52 |

L100–L70: *Lentinula edodes* (strain L5), with 100%–70% mushroom and 0%–30% ground yellow corn. P100–P70: *Pleurotus ostreatus* (strain Po-IAP), with 100%–70% mushroom and 0%–30% ground yellow corn CP: crude protein; DF: acid detergent fiber, NDF: neutral detergent fiber; ADF: acid detergent fiber; Cel: cellulose, H: hemicellulose, L: lignin. The values are the mean, n = 8. SEM: standard error of the mean. Means in a column without a common letter differ from each other, p < 0.05. Differences were determined by one-way ANOVA; Tukey's test was used as a post-hoc test.

### Determination of mycotoxin by thin layer chromatography (TLC)

The analysis of mycotoxin by thin layer chromatography demonstrated that none of the micro silages presentd contamination by Aflatoxin B1, Zearalenone and Ochratoxin A (Fig 1).

### Quantification of total phenolic compounds and tannins of micro-silages

The phenolic content of micro-silages formulated with different levels of SMS from *Lentinula edodes* (strain L5) and *Pleurotus ostreatus* (strain Po-IAP) is shown in Fig 2A. The highest concentration of polyphenols ($P = 0.001$) was in L100 (9.0 mg GAE/ g DM), followed by L90, L80 and L70 (8.6, 8.2 and 8.2 mg GAE/ g DM, respectively); meanwhile the lowest

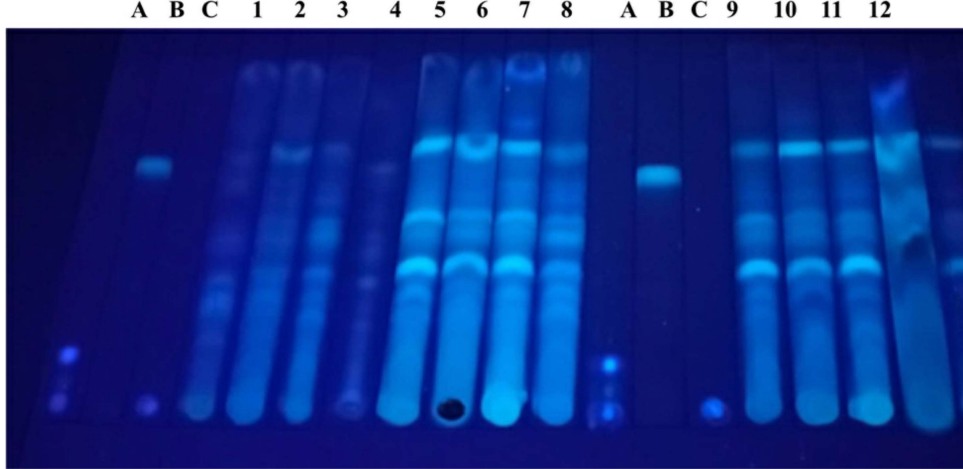

**Fig 1. Mycotoxin determination by thin layer chromatography (TLC).** Lane A: standard of Aflatoxin B1, lane B: standard of Zearalenone, lane C Ochratoxin A, lane 1: corn stover, lane 2: non inoculated mushroom substrate, lane 3: SMS of *Lentinula edodes (strain L5)*, lane 4: SMS of *Pleurotus ostreatus* (strain Po-IAP), lane 5: P100, lane 6: P90, lane 7: P80, lane 8: P70, lane 9: L100, lane 10: L90, lane 11: L80, lane 12 L70.

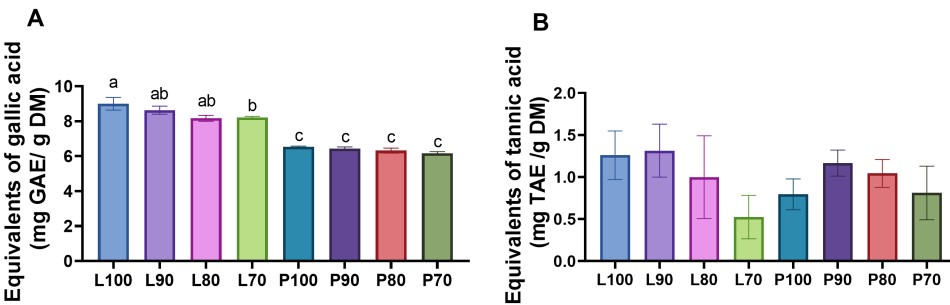

**Fig 2. (A) Phenolic and (B) tannin content of micro-silage from SMS of Lentinula edodes (strain L5) and Pleurotus ostreatus (strain Po-IAP).** Values are means±SEM, n=3. Labeled means without a common letter differ, *P*<0.05, a>b>c.

concentration of polyphenols (*P*=0.001) was observed in all the *Pleurotus ostreatus* (strain Po-IAP) silage P100 (6.5 mg GAE/ g DM), P90 (6.4 mg GAE/ g DM), P80 (6.3 mg GAE/ g DM) and P70 (6.1 mg GAE/ g DM). No difference (*P*=0.642) was observed in the tannin content (Fig 2B).

### In vitro rumen fermentation

A significant difference (*P*=0.0096) was observed in the in vitro digestibility of dry matter (IVDMD) due to the interaction between the inclusion level and the SMS (Fig 3). Silages containing 70% inclusion of SMS (L70 and P70) exhibited higher digestibility in comparison to those made with 100% SMS (L100 and P100), showing a percentage difference of up to 20.6% between the two formulations. The main effect of inclusion level and SMS strain, as well as their interaction, are reported in S2 Table. Inclusion level had a significant effect on IVDMD (*P*<0.0001), with lower inclusion (70%) resulted in higher IVDMD, while the main effect of fungal strain was not significant (*P*=0.4642).

### Determination of ammoniacal nitrogen and pH

No differences in the pH level were observed after *in vitro* fermentation due to the interaction between the inclusion level and the SMS (*P*=0.8954). However, a significant pH decrease (*P*<0.0001) was observed in treatments with SMS of *Pleurotus ostreatus* (strain Po-IAP) in comparison with treatments with *Lentinula edodes* (strain L5). Additionally, treatments with higher inclusion of SMS showed the lower (*P*<0.0001) pH concentration (S2 Table). The pH range after IVDMD varied from 6.72 for the formulation containing 70% SMS of *Pleurotus ostreatus* (strain Po-IAP) (P70) to 7.08 in the 100% SMS of *Lentinula edodes* (strain L5) formulation (Fig 4A).

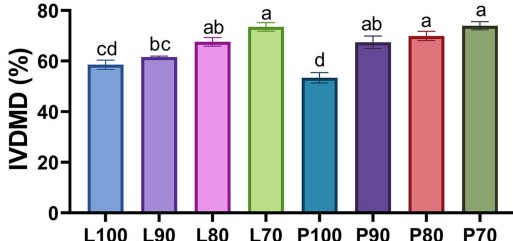

**Fig 3. In *vitro* digestibility of dry matter of micro-silage from SMS of *Lentinula edodes* (strain L5) and *Pleurotus ostreatus* (strain Po-IAP).** Values are means±SEM, n=60 Labeled means without a common letter differ, *P*<0.05, a>b>c.

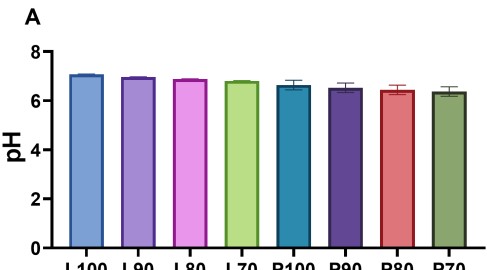 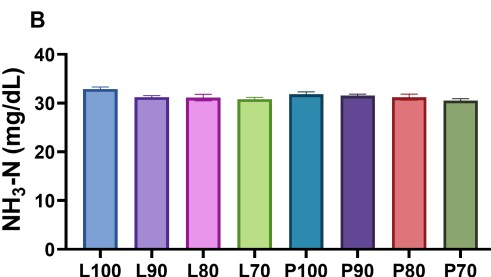

**Fig 4. (A) pH value and (B) ammoniacal nitrogen (NH₃-N) after *in vitro* fermentation of micro-silages from SMS of *Lentinula edodes* (strain L5) and *Pleurotus ostreatus* (strain Po-IAP). Values are means ± SEM, n = 60.**

Similarly, the interaction between SMS strain and inclusion level, had not significant effects ($P = 0.2924$) on NH₃-N (Fig 4B), with the lowest (30.5 mg/dL) and highest (32.9 mg/dL) values recorded in P70 and L100, respectively. No effect was observed for the SMS ($P = 0.3785$). A significant decrease ($P < 0.0001$) was observed as the inclusion level diminished (S2 Table).

### Ruminal kinetics

As observed in Fig 5B and 5D, there was an interaction between SMS strain and inclusion for maximum gas production (Vmax) and lag phase (L) ($P < 0.001$). Independently the strain, *Pleurotus ostreatus* (strain Po-IAP) or *Lentinula edodes* (strain L5), the silage formulated with 100% SMS inclusion produced a lower Vmax (P100: 113.3 mL/g DM; L100: 140.8 mL/g DM) than those made with 70% SMS (P70: 274.4 mL/g DM; L70: 306.5 mL/g DM). The formulation that presented the highest Vmax was L70 (Fig 5B), corresponding to the silage made with 70% SMS of the *Lentinula edodes* (strain L5). It can also be observed that the *Lentinula edodes* (strain L5) SMS silages produced more gas ($P < 0.05$) than those made with *Pleurotus ostreatus* (strain Po-AIP) SMS. The main effects of Vmax and L are shown in S3 Table.

No statistically significant differences ($P = 0.1726$) were observed in the gas production rate (S), for the interaction between SMS strain and inclusion level (Fig 5C). However, as the inclusion level increased a lower ($P < 0.0001$) gas production rate (S) was observed. In the same way, there was an effect due to SMS strain used, silages formulated with SMS from *Lentinula edodes* (strain L5) showed a higher ($P = 0.0116$) gas production rate (S) in comparison with silages of SMS from *Pleurotus ostreatus* (strain Po-IAP) (S3 Table).

### Determination of volatile fatty acid (VFA)

In the case of VFA production, no significant differences ($P = 0.2645$) were due to the interaction between SMS strain and inclusion level observed between treatments (Fig 6A). However, there is a difference ($P < 0.001$) due to the inclusion level, where silages prepared with lower level of inclusion (70%) showed the higher VFA production (S3 Table). No differences ($P = 0.3126$) were observed due to the SMS strain (S4 Table).

There were no significant differences in the production of acetic (Fig 6B) ($P = 0.0655$) and butyric acid (Fig 6D) ($P = 0.1879$) production for the interaction of SMS strain and inclusion level. In the case of acetic acid and butyric acid, it was observed that decreasing the inclusion level caused a reduction in its concentration ($P < 0.0001$ for both acids) (S4 Table). No significant differences were observed due to SMS strain on the production of acetic acid ($P = 0.3126$) or butyric acid $P = 0.0638$) (S4 Table).

Meanwhile, in the case of propionic acid production (Fig 6C), significantly higher ($P = 0.0414$) production was shown by treatments L90, L80 and P80, reaching 25.01%, 24.95%, and 24.87%, respectively. The main effects of propionic acid are shown in S4 Table.

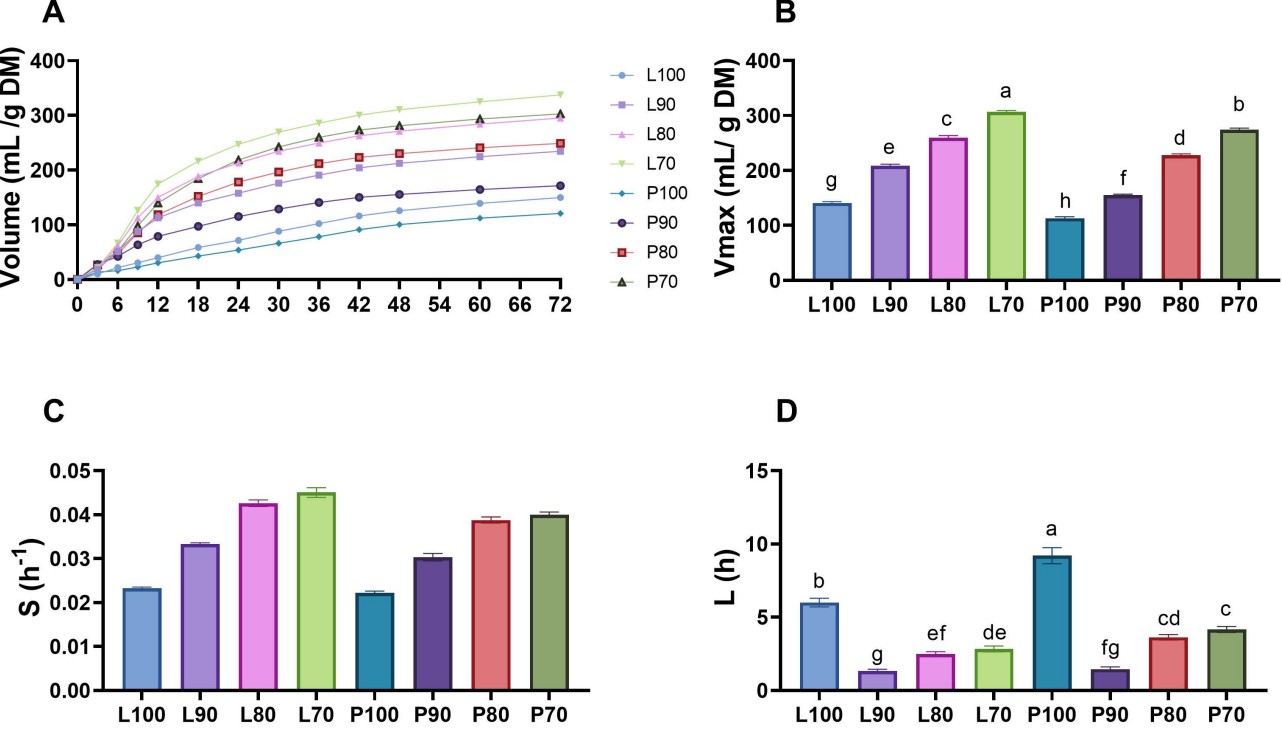

**Fig 5. (A) Gas production, (B) maximum volume of gas produced (Vmax), (C) gas production rate (S), (D) lag phase (L) after *in vitro* fermentation of micro-silages from SMS of *Lentinula edodes* (strain L5) and *Pleurotus ostreatus* (strain Po-IAP).** Values are means ± SEM, n = 60. Labeled means without a common letter differ, $P < 0.05$, a > b > c.

## Discussion

The present study demonstrated that ensiling of SMS from *Pleurotus ostreatus* (strain Po-IAP) and *Lentinula edodes* (strain L5) with an adequate carbohydrate source such as ground yellow corn is a viable alternative for the use of this agro-industrial waste as an animal feed, because it yields good quality silage with acceptable organoleptic characteristics and good nutritional quality with better *in vitro* digestibility and fermentation kinetics. Further *in vivo* studies are encouraged to validate the *in vitro* results and evaluate their impact on animal performance and metabolism.

The ever-growing demand for meat and dairy products has increased the demand of conventional forage crops for animal feed. As previously stated, one common practice of preserving these crops is through ensilaging. For instance, a widely used crop is the maize; however, its production is associated with intensive land use, biodiversity loss, and significant environmental impacts such as soil erosion [39]. Therefore, it has been an increase in the study of non-conventional sources of silage that not only enhances animal nutrition but also encourages circular economy. In this regard, a residue that has gained importance is the SMS, a residue generated from the cultivation of edible mushrooms.

When using non-conventional sources of silage, factors such as organoleptic characteristics should be considered, as they serve as a qualitative indicator of its quality. In the present study we considered the color, odor, the presence of fungal spots, particle size and spots between layers as parameters of the quality of the silages. It has been well established that the color of silage is mainly determined by factors such as, the color of the material to be ensilated [40] and the fermentation process. During the fermentation process, temperature increases as microorganisms break down carbohydrates releasing energy in the form of heat, as well as $H_2O$ and $CO_2$ [41]. Prolonged respiration causes a greater temperature increase, which damages the color of the final product, mainly causing dark coloring [42]. In the present study most of

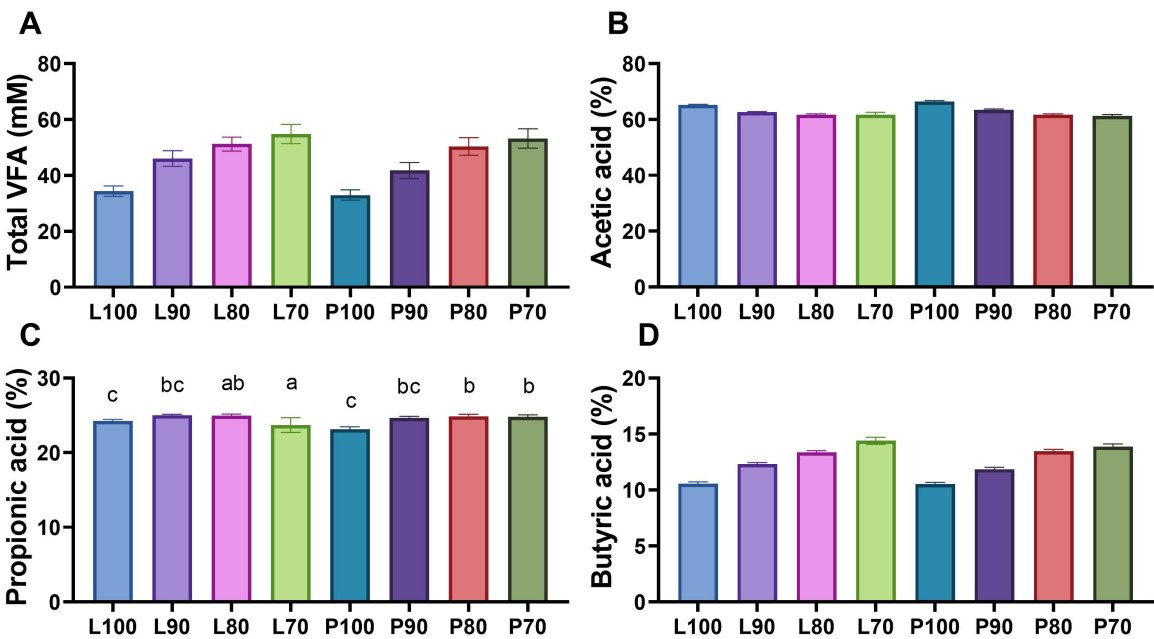

**Fig 6. (A) Total volatile fatty acid (VFA) concentration, (B) acetic acid (%), (C) propionic acid (%) and (D) butyric acid (%) after *in vitro* fermentation of micro-silage from SMS of *Lentinula edodes* (strain L5) and *Pleurotus ostreatus* (strain Po-IAP).** Values are means ± SEM, n = 20. Labeled means without a common letter differ, $P < 0.05$, a > b > c.

the silages presented an acceptable light brown color (Table 3) suggesting good fermentation and an adequate compaction process of the material in the silo [40].

The silage particle size is an important factor in silage quality, because excessively long particles increase the chewing required to swallow the feed bolus, leading to longer rumination time, but shorter consumption time [43], which reduce dry matter intake (DMI). Kononoff *et al.* (2003) observed that as particle size decreased, DMI increased linearly [44]. Similarly, Kononoff and Heinrichs (2003) compared the effects of alfalfa with a mean geometric length ranging from 4.1 to 6.8 mm and found that daily feeding time increased by 36 min per day as silage particle size increased [45]. At the same time, DMI decreased by 3.3 kg/day. In the present study, the homogeneous particle size (1–2.5 cm; Table 3) obtained indicates that the silage could be used as an animal feed without affecting DMI, it is however necessary to perform further studies to evaluate its impact on animal performance.

The odor of the ensiled product is an indicator of physical quality and is closely related to the fermentation process [40]. In this study, the odor of the micro-silages was rated as adequate for those made with the *Pleurotus ostreatus* (strain Po-IAP), while those made with *Lentinula edodes* (strain L5) received an inadequate rating (Table 3). Good silage has an acidic but not spicy aroma. This is because the bacteria that develop during an adequate fermentation process produce lactic acid [46]. In addition, a distinct aroma is a well-known characteristic of mushrooms from the *Lentinula* spp., family [47], suggesting that the odor detected in this silage was primarily associated with the fungal strain rather than an improper ensiling process.

Muck et al. (2018) emphasize that a low pH in silages is crucial for their stability and conservation, as it indicates efficient fermentation [48]. They explain that lactic acid production is the main factor responsible for the reduction of pH, which inhibits the growth of undesirable microorganisms that could deteriorate the silage. Additionally, a lower pH extends silage shelf life by preserving nutrients [49]. In the present study, the treatments that presented the lowest pH were L70 and P70 (3.9; Table 3), which is below that reported by various authors for silages made from alternative raw materials

to corn [40,50,51], while the L100 and P100 formulations presented a higher pH (5.1 and 5.3, respectively; Table 3) than those reported by the same authors. Tahuk et al. (2020) consider silages with a pH lower than 4.4 to be of good quality, whereas those with a pH of 4.8 or higher are classified as poor quality [40]. They also associate high pH with the low content of soluble carbohydrates in the material to be ensiled. Another factor influencing pH is the availability of lactic acid-producing bacteria in the silage [49]. Lactic acid has the highest acidity compared to other organic acids formed during fermentation [48].

Finally, the absence of fungi indicates that the fermentation process was optimal. As previously stated, the acidic pH produced by lactic acid bacteria can prevent the growth of undesirable fungi [40,42]. In this regard, based on the SBL, the micro-silages demonstrated adequate quality, except for formulation P100, which was rated as average quality. This finding is consistent with P100 being the formulation with the highest pH.

Even though it has been demonstrated that SMS is a suitable substrate for silage [52,53], the present study is among the few that assessed their organoleptic characteristics. To fully demonstrate that silage from SMS has good characteristics it is highly recommended to include a larger number of trained judges and standardized organoleptic analysis. However, taking notice of the odor detected in the *L. edodes* silage, it is necessary to perform palatability trials in ruminants to evaluate feed intake, animal preference and impact on animal performance.

In addition to the organoleptic characteristics, it is necessary to have good nutritional properties to ensure good use by animals. In the present study, there was a decrease in CP content as the inclusion percentage of SMS was reduced for both strains (Table 4). This indicates that SMS was the primary protein source in silage. This behavior can be explained by that the spent substrates used for production of fungal fruit bodies were colonized by mycelium, which contributes a significant amount of protein [7].

However, the CP content reported for this type of SMS by different authors [7,54] is not as high as the values obtained in this study. These variations may be attributed to differences in mushroom cultivation processes, environmental conditions during the mycelium invasion and fruiting stages, and the degree to which the substrates were colonized and depleted.

One advantage of SMS silages is that most of them exceed the CP content provided by corn silage (7.86%), which is the most widely used ingredient in ruminant nutrition worldwide and is obtained through lactic fermentation [55].

On the other hand, ash values decrease as the SMS inclusion percentage is reduced with both strains (Table 4). Additionally, silages made with *Pleurotus ostreatus* (strain Po-IAP) SMS showed higher values than those made with *Lentinula edodes* (strain L5) SMS. In this regard, Lorenzana *et al*. (2023) reported a significant increase ($p < 0.05$) in the mineral content of spent substrate from three different strains of white-rot fungi compared to non-colonized substrate, including *Lentinula edodes (strain L5)* [7]. Based on these findings, we can infer that the increase is also a consequence of substrate colonization by fungal mycelium.

The content of NDF, CP, digestibility, and particle size, along with fermentation end-products, influence feeding behavior, ruminal passage rate, dry matter intake (DMI), and consequently, milk production efficiency [56,57]. Numerous studies indicate that diets with higher NDF content, forages with lower NDF digestibility, and larger particle sizes increase the time required for feed consumption. Increased chewing time during feeding and rumination can be a major limiting factor for dry matter intake in high-producing dairy cows [43]. Additionally, relatively high ADF content in forage could reduce dry matter digestibility [56].

In this context, a decrease in NDF, ADF, cellulose, and lignin content is observed as SMS inclusion is reduced in the micro-silages, with the lowest values recorded in formulations containing 70% SMS for both strains (Table 4). Lorenzana *et al*. (2023) reported values of 45.4% NDF and 38.1% ADF for unensiled *Lentinula edodes* (strain L5) SMS and 62.1% NDF and 41.9% ADF for *Pleurotus ostreatus* (strain Po-IAP) SMS [7], all of which are higher than those found in this study for silages with 70% SMS. This could be explained by the fact that the ensiling process partially degrades the fibrous structures of forages, thereby increasing nutrient digestibility [48]. For example, *Pleurotus ostreatus* produces enzymes

such as endoglucanase that degrade cellulose into glucose and oligosaccharides. This cellulolytic enzyme may degrade cellulose during storage and consequently reduce NDF and ADF and increase non-fibrous carbohydrates (NFC) concentration [58]. This aligns with the findings of this study, as the L70 and P70 formulations exhibited the highest IVDMD.

The decrease in ADF for both strains is also associated with the reduction in cellulose and lignin values as SMS inclusion decreases in the formulations. Additionally, it is influenced by the low lignin content contributed by corn, which is in range 1.24–2.33% [59], compared to the lignin content of spent substrates 15.3% in *Lentinula edodes* (strain L5) and 12.1% in *Pleurotus ostreatus* (strain Po-IAP) [7].

Therefore, silages formulated with 70% SMS, due to its nutritional value, could promote higher feed intake and greater digestibility compared to unensiled SMS and the other silage formulations evaluated in this study. However, further studies are needed to fully elucidate the effect of this formulation and animal performance.

A potential disadvantage of silages elaborated with agro – industrial residues, which involves a certain storage time in humid conditions at room temperature, is the presence of mycotoxins. However, none of the micro-silages made in this experiment showed contamination by Aflatoxin B1, Zearalenone and Ochratoxin A (Fig 1). In this sense, García *et al.* (2024) reported levels of mycotoxins in fruit and vegetable silages below the corresponding detection limits, despite having observed molds in some fruits and vegetables during processing, they mention that this is because the proper silage fermentation process controlled their development [50]. It has been reported that some lactic acid bacteria and propionic acid-producing bacteria can sequester various mycotoxins and have the potential to degrade these compounds [60]. Indicating that good silage conditions are critical in the control of mycotoxins.

Another characteristic that has gained increased interest in recent years is the presence of bioactive compounds in animals feed, it has been demonstrated the agro-industrial residues have important levels of bioactive compounds, such as polyphenols, that might improve animal health, fermentative parameters and improve the quality of both milk and meat [4,61].

Our results showed that the concentration of polyphenols was higher in the *Lentinula edodes* (strain L5) (8.2–9.0 mg GAE/ g DM) silages in comparison to *Pleurotus ostreatus* (strain Po-IAP) silages (6.1–6.5 mg GAE/ g DM), which is consistent with previous studies that demonstrate that *Lentinula edodes* have a higher content of phenolic compounds in comparison to other edible mushrooms [62]. For instance, it has been reported that aqueous extracts from the fruiting body of *Lentinula edodes* (strain L5) contain 24.14 mg GAE/g extract, whereas extracts from *Pleurotus ostreatus (strain Po-IAP)* have 19.37 mg GAE/g extract [63].

The difference in phenolic content between these strains of mushrooms may be associated with the ability of *Lentinula edodes* to metabolize lignin. It has been shown that a content of 0.10% of lignin in the substrate for the growth of *Lentinula edodes* enhances mycelia growth and promote the accumulation of phenolic compounds [64]. In the present study, the substrate used for the mushroom production was formulated with 80% corn stover [7], which is rich in lignin. This could explain the higher content of these compounds in the silage from *Lentinula edodes* (strain L5).

Polyphenolic compounds, which are secondary metabolites of plants, have been extensively studied and shown to exert a wide range of beneficial effects in animals, including improved performance, modulation of gut microflora, enhancement of the immune system, stress mitigation, and regulation of gene expression through nutrigenomics. These compounds possess notable antioxidant, antimicrobial, anti-inflammatory, anti-allergic, antimutagenic, and immunomodulatory properties [65,66]. In this regard, Safari et al. (2018) [67] reported an improvement in milk yield and composition (3.5% fat-corrected milk, fat content, and total solids) when using pomegranate seed pulp, which contained higher levels of polyphenols compared to pomegranate seed (25.5 vs. 1.61 g/kg, respectively). These findings suggest that higher polyphenol content, as observed in the silages of the present study, particularly those made with SMS from Lentinula edodes, could improve productive performance or certain characteristics of the final product in animal feed. Similarly, Bonanno et al. (2019) [68] reported a higher antioxidant capacity in sheep fed with 20% mushroom myceliated grains (MMGs), reaching 17.83 mmol Trolox equivalents/kg dry matter (DM), compared to 9.97 and 9.18 mmol Trolox equivalents/kg DM in

animals fed with 10% and 0% MMGs, respectively (P<0.001). This increase suggests greater oxidative stability of cheese fat and a potential enrichment of the cheese with antioxidant compounds, an effect attributed to the bioactive compounds, such as polyphenols, present in the MMGs.

On the other hand, mushroom tannin content is very low. For instance, in *Pleurotus ostreatus*, it has been reported to a range between 6.31–25.12 mg/100g [69], which is relatively low in comparison to other feed products. *Pleurotus ostreatus* SMS have ligninolytic enzymes like laccase, lignin peroxidase and manganese peroxidase, which are capable of degrading phenolic and non-phenolic compounds [70]. Therefore, it was expected that the silage from SMS would have a low content of tannins and would not have a detrimental effect on the animal health.

The digestibility of forage is one of the most crucial indexes which could directly reflect the quality of silage. The IVDMD observed in the silages with the highest digestibility (P70=74%, L70=73.5%; Fig 3) was comparable to the value reported by Pozo *et al.* (2021) for corn silage (73.8% IVDMD) [71], which is considered high quality when the ensiling process is performed optimally [71]. Moreover, these values exceed those reported by Jančík *et al*. (2022), indicating that silages formulated with 70% SMS and 30% corn can be classified as high-quality forages [72].

Guerrero *et al.* (2023) reported a minimum IVDMD of 47.3% for corn stover [73], while González *et al.* (2017) recorded 67.51% for ground corn [74]. These findings support the hypothesis that increasing the proportion of corn in silage formulations enhances *in vitro* digestibility. Similarly, the P70, L70, and P80 treatments exhibited higher IVDMD percentages compared to the values reported for both untreated corn and stover. Numerous studies have demonstrated that the ensiling process improves forage digestibility by partially breaking down fiber through effective fermentation, which optimizes feed quality, preserves nutritional properties, and increases nutrient availability [48].

IVDMD was similar for both strains; however, Lorenzana *et al.* (2023) reported a higher digestibility in the SMS of *Lentinula* spp., compared to the SMS of *Pleurotus* spp [7]. This difference may be attributed to modifications induced by the ensiling process, as noted by Muck *et al.* (2018), who emphasized the role of fermentation in enhancing the digestibility of substrates used in animal feed [48].

The improvement in digestibility is attributed to the reduction in the percentage of spent mushroom substrates (SMS) in the silages and the increase in corn concentration. This notable difference underscores the importance of incorporating a source of soluble carbohydrates in the formulation of these silages to enhance their nutritional quality and digestibility.

According to Dijkstra *et al.* (2012), the optimal pH of the rumen ranges from 6.0 to 7.0, within which the ruminal microbiota functions efficiently, promoting fiber digestion [75]. In addition, they point out that a drastic drop in pH can indicate a high acidogenic potential, which can cause ruminal acidosis, a condition that negatively affects the animal's health and compromises nutrient digestibility. In this context, the pH values observed in the *in vitro* fermentation ranged from 6.72 to 7.08 across the different formulations (Fig 3A and 3B), suggesting that the silages in this study create conditions conducive to maintaining an optimal ruminal pH.

In the rumen, protein and non-protein nitrogen are broken down into ammonia-N, which microbes can utilize along with peptides and amino acids to synthesize microbial proteins. Therefore, ammonia-N could somewhat reflect the growth rate and population of microbial cells [76]. Rodríguez *et al.* (2007) mention that feeding diets should maintain $NH_3$-N concentrations between 4 and 10.0 mg of $NH_3$-N/100 mL of rumen fluid [76,77], for microbial protein synthesis, as well as for providing available energy to the ruminal ecosystem. Ammonia values from all formulations exceed 30.5 mg/dL, which is higher than the levels mentioned by Rodriguez *et al.* (2007) [77]. These results are also greater than those reported by Paula *et al.* (2017) for soybean meal (15.7 mg/dL of $NH_3$-N), a protein source considered highly degradable [78]. This suggests a high protein degradation of the SMS silages used in this study. A possible explanation is that the ensiling process improves nitrogen utilization efficiency in ruminants, increasing $NH_3$-N availability and promoting microbial protein synthesis in the rumen [49].

Knowles *et al.* (2017) mention that the secondary metabolites present in certain forages, such as legumes, can influence protein degradation and synthesis in the rumen [79]. This effect can occur either by directly impacting ruminal

microorganisms or by interacting with nutrients, depending on the type and quantity of the metabolite. In this study, the lower presence of secondary metabolites due to the ensiling process did not affect the *in vitro* protein degradation.

The in vitro gas production technique can be used to assess the nutritive value of roughages and to identify differences in their potential digestibility and energy content. Chemical composition and in vitro digestibility are useful indicators for the preliminary evaluation of the potential nutritive value of previously uninvestigated ingredients [80,81]. The formulations that achieved the highest Vmax were L70 and P70 (306.5 and 274.4 mL/g DM, respectively), which is directly related to IVDMD, as these formulations also exhibited the highest digestibility. In this context, some authors have reported gas production values at 72 hours ranging from 226.82 to 360 mL/g DM for corn silage, which is considered the most used silage product worldwide. The values obtained in this study for treatments P80, L80, P70, and L70 (248.73, 295.20, 302.92, and 337.62 mL/g, respectively) fall within that range, indicating that SMS silage in these mixtures produces an in vitro fermentation pattern very similar to that of corn silage, and therefore, is suitable for animal feeding [82,83].

Similarly, the treatments with the lowest digestibility also had the lowest Vmax (P100 = 113.3 and L100 = 140.8 mL/g DM; Fig 3). In this regard, Velásquez *et al*. (2013) mention that gas volume production and dry matter degradability have a positive correlation, meaning that as feed digestibility increases, gas production also rises [84].

This could be attributed to the increased supply of NFC from ground corn, as these were the treatments with the highest inclusion levels. A higher energy density promotes rapid bacterial growth, increasing volatile fatty acid production and, consequently, gas volume [85]. Additionally, ground corn has a much larger surface area compared to SMS, allowing ruminal microorganisms greater access to starch molecules, leading to faster fermentation of these carbohydrates and, therefore, higher gas production volumes [84].

VFAs are the predominant end products of the rumen and are the main metabolizable energy for ruminants. Production and the proportions of VFA can somewhat reflect the metabolism status of rumen microorganisms to let us estimate whether the microbiota in the rumen is predominantly fibrolytic or predominantly amylolytic [43,86] generally, a greater extent of substrate degradation was accompanied by more VFA production. This is consistent with the findings of this study, as some of the formulations with higher digestibility also showed higher propionic acid production (Fig 4B). In addition, a higher concentration of NFC is rapidly assimilated by ruminal microorganisms and has a positive correlation with VFA syntheses [58], what could be happening in silages with lower SMS inclusion. The total VFAs concentration in this study is lower (Fig 6) than that reported by Baek *et al*. (2017) for SMS silages of *Pleurotus ostreatus* (77.2 mM – 103.7 mM) [54]. However, this difference is not due to lower digestibility but rather to differences in sampling times during fermentation, as well as the use of fermentation in their study. Similar to what was reported in this work (supp Table 3), Zhang *et al*. (2024) report a higher amount of VFAs for *Lentinula edodes* (7219 mg/L) compared to *Pleurotus ostreatus* (6358 mg/L) and associate it with changes in the ruminal microbiota, however, they mention that more research is needed [87]. The increase in the total VFA concentration is due to the higher NFC content provided by the ground corn in the L70 and P70 formulations [78].

The higher proportion of propionic acid in formulations with a grater inclusion of ground corn (Fig 6) may be an indicator of adequate lactic acid production during ensiling. Kung *et al*. (2018) reported that a higher lactate content in silages leads to a greater proportion of ruminal propionic acid [49]. Likewise, this may also suggest an increase in the populations of Genus *Prevotella*. Chen *et al*. (2019) mention that propionate is the main fermentation product of this bacterial genus and associate the increase in the molar proportion of propionic acid with a rise in *Prevotella* populations [86].

Propionic acid is the only major gluconeogenic VFA in the rumen. It can help to improve the energy balance and decrease the risk for metabolic disorders of cows in early lactation [86]. An important consideration when increasing propionic acid in ruminal fermentation is that silage starch content and fermentability may influence ruminal propionate production and thereby exert substantial control over meal patterns and feed consumption [43]. Allen *et al*. (2009) mention that one mechanism by which ruminants DMI is through high-starch diets, which increase propionic acid production in the rumen [88]. This leads to a rise in VFA concentrations in the bloodstream, particularly propionate, which is converted into glucose and enters circulation. As a result, this triggers a satiety effect, potentially reducing DMI.

Lu *et al.* (2022), indicated that the synthesis of acetate resulted in a greater volume of gas production than propionate synthesis. In the present study, no significant difference was observed in acetic acid [80]. However, we observed a tendency to present a higher proportion of acetic acid in the formulations with 100% SMS (p = 0.06 inclusion*strain interaction, Fig 6), this coincides with the micro-silages with a higher amount of NDF, ADF, Cel and L. Acetic acid formation mainly results from structural carbohydrate fermentation by cellulolytic bacteria (*R. flavefaciens*, *R. albus*, *F. succinogenes* and *B. fibrisolvens*) in the rumen [Chen et al., 2019]. Moderate concentrations of acetic acid silage can be beneficial because they inhibit yeasts, resulting in improve stability when silage is exposed to air [49].

Overall, the results of the present study indicate that silage formulated with 70% SMS, might be beneficial in different feeding systems. Furthermore, their use align with models of circular economy, in which, one of the main objectives is reducing waste generation and enhancing the use of resources by repurposing by-products into valuable inputs for other systems [1]. Silages formulated with SMS form *Pleurotus ostreatus*, showed better organoleptic characteristics and had fermentation parameters comparable to high quality silages. Therefore, silage from SMS of *Pleurotus ostreatus* are nutritionally viable but also applicable and could be used as a sustainable alternative to waste management.

## Conclusions

Of the eight silage formulations evaluated in the present study, those formulated with 70% inclusion of spent mushroom substrate (SMS) showed higher IVDMD and fermentative parameters. However, the silage formulated with *Lentinula edodes (strain L5)* exhibited an undesirable odor that could affect animal acceptance *in vivo*. Our results demonstrate that SMS silage is a viable option for using a by-product generated from a growing industry. Further research is recommended to evaluate the impact of *in vivo* use of SMS silage on animal health, intake behavior and productive parameters.

## Supporting information

**S1 Raw image. Mycotoxin determination by thin layer chromatography (TLC).** Picture was taken using a cellphone camera (iPhone model 14) with a handheld Mineralight® Multi-D Lamp, UVSL-25, corresponding to Fig 2.
(PDF)

**S2 Table. In vitro digestibility of dry matter of micro-silages, pH value and ammoniacal nitrogen (NH$_3$-N) from spent mushroom substrates (SMS) of *Lentinula edodes* L5 and *Pleurotus ostreatus* IAP.**
(DOCX)

**S3 Table. Maximum volume of gas produced (Vmax), gas production rate (S), and lag phase (L) after in vitro fermentation of micro-silages from SMS of *Lentinula edodes* L5 and *Pleurotus ostreatus* IAP.**
(DOCX)

**S4 Table. Total volatile fatty acid (VFA) concentration, acetic acid (%), propionic acid (%), and butyric acid (%) after in vitro fermentation of micro-silages from SMS of *Lentinula edodes* L5 and *Pleurotus ostreatus* IAP.**
(DOCX)

## Author contributions

**Conceptualization:** Angélica Valeria Lorenzana-Moreno, Claudia C. Márquez-Mota.

**Data curation:** Angélica Valeria Lorenzana-Moreno, José Moisés Talamantes-Gómez, Claudia C. Márquez-Mota.

**Formal analysis:** Angélica Valeria Lorenzana-Moreno, Diana Victoria Valdés-Meléndez, Hermilo Leal Lara, Claudia C. Márquez-Mota.

**Funding acquisition:** Claudia C. Márquez-Mota.

**Investigation:** Angélica Valeria Lorenzana-Moreno, Diana Victoria Valdés-Meléndez.

**Methodology:** Angélica Valeria Lorenzana-Moreno, José Moisés Talamantes-Gómez, Augusto César Lizarazo-Chaparro, Claudia C. Márquez-Mota.

**Project administration:** Claudia C. Márquez-Mota.

**Resources:** Hermilo Leal Lara.

**Supervision:** Angélica Valeria Lorenzana-Moreno, Augusto César Lizarazo-Chaparro, Claudia C. Márquez-Mota.

**Validation:** Angélica Valeria Lorenzana-Moreno.

**Visualization:** Angélica Valeria Lorenzana-Moreno, Claudia C. Márquez-Mota.

**Writing – original draft:** Angélica Valeria Lorenzana-Moreno, Claudia C. Márquez-Mota.

**Writing – review & editing:** Angélica Valeria Lorenzana-Moreno, Claudia C. Márquez-Mota.

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
