## [Decision Letter · Decision Letter 0]

15 May 2025

PONE-D-25-12392Formulation of silages from spent mushroom substrates of Pleurotus ostreatus and Lentinula edodes: organoleptic properties, phenolic content, in vitro digestibility, gas production and ruminal kineticsPLOS ONE

Dear Dr. Márquez-Mota,

Thank you for submitting your manuscript to PLOS ONE. After careful consideration, we feel that it has merit but does not fully meet PLOS ONE’s publication criteria as it currently stands. Therefore, we invite you to submit a revised version of the manuscript that addresses the points raised during the review process.

We look forward to receiving your revised manuscript.

Kind regards,

Agung Irawan

Academic Editor

PLOS ONE

Journal Requirements:

2. Thank you for stating the following financial disclosure: [This research was financially supported by grant: PAPIIT IN212822, from the Dirección General de Asuntos del Personal Académico (DGAPA)- UNAM.]. 

3. Thank you for stating the following in the Acknowledgments Section of your manuscript: [This research was financially supported by grant: PAPIIT IN212822, from the Dirección General de Asuntos del Personal Académico (DGAPA)- UNAM.]

Please remove any funding-related text from the manuscript and let us know how you would like to update your Funding Statement. Currently, your Funding Statement reads as follows: [This research was financially supported by grant: PAPIIT IN212822, from the Dirección General de Asuntos del Personal Académico (DGAPA)- UNAM.]. 

4. Please provide a complete Data Availability Statement in the submission form, ensuring you include all necessary access information. If your research concerns only data provided within your submission, please write "All data are in the manuscript and/or supporting information files" as your Data Availability Statement.

Additional Editor Comments:

I agree with 2 reviewers that this manuscript requires substantial revisions before can be re-considered for publication. Please carefully address the issues raised by the reviewers by following journal's instruction. 

Reviewers' comments:

Reviewer's Responses to Questions

**Comments to the Author**

1. Is the manuscript technically sound, and do the data support the conclusions?

Reviewer #1: Yes

Reviewer #2: Partly

2. Has the statistical analysis been performed appropriately and rigorously? 

Reviewer #1: Yes

Reviewer #2: No

3. Have the authors made all data underlying the findings in their manuscript fully available?

Reviewer #1: Yes

Reviewer #2: Yes

4. Is the manuscript presented in an intelligible fashion and written in standard English?

Reviewer #1: Yes

Reviewer #2: Yes

5. Review Comments to the Author

Reviewer #1: The paper, rather grandly titled “Formulation of silages from spent mushroom substrates of Pleurotus ostreatus and Lentinula edodes: organoleptic properties, phenolic content, in vitro digestibility, gas production and ruminal kinetics,” tackles what I reckon is a pretty crucial and timely area. I mean, sustainable livestock feed? Definitely something we need to be thinking about, isn't it? So, yeah, the whole subject matter really grabbed my attention, and I settled in to read the manuscript with genuine interest. It certainly seems to snuggly fit within the journal's usual scope, which is a plus.

However – and you knew there was a 'however' coming, right? – I must say that in its present iteration, the paper does seem to have a few... shall we say, areas that could do with a bit more polish. Nothing major, mind you, but a few things that sort of stuck out.

Abstract

• is generally clear and concise, providing a good overview of the study. However, the phrase "lower production cost has increased significantly" sounds a little awkward. Perhaps rephrasing it to something like "the search for alternative feed sources to lower production costs has increased significantly" would flow better.

• It might be helpful to briefly mention the specific percentages of SMS and corn used in the formulations within the abstract to give the reader a quicker grasp of the experimental design.

Introduction

• While does a decent job of setting the stage by discussing the importance of sustainable food production and the potential of spent mushroom substrate (SMS), it feels a bit... rushed, perhaps? It might benefit from a slightly more in-depth exploration of the existing challenges in livestock feed and how SMS can really address them. For instance, elaborating on the environmental and economic benefits a bit more could strengthen the intro. Also, the flow could be smoother; at times, it jumps a bit abruptly between ideas.

• hypothesis is clearly stated, which is good! However, the justification for focusing on Pleurotus ostreatus and Lentinula edodes specifically could be stronger. Why these two? Are they particularly promising, or were they simply the ones available for the study? Clarifying this would add more context.

• when discussing circular economy models, it might be beneficial to briefly explain what these models entail for readers who may not be familiar with the concept in this context. Just a sentence or two could suffice.

• use of "(strain Po-IAP)" and "(strain L5)" after mentioning Pleurotus ostreatus and Lentinula edodes is good for clarity. Ensure this consistent throughout the manuscript.

• Line 45 I suggest citing 10.1080/19440049.2024.2414954; 10.1038/s41598-025-00675-2 and 10.3389/fvets.2024.1441905 to strength the circular feed sentence

Materials and Methods

• description of the animal management and treatments is adequate, and it's good to see the ethics approval mentioned. However, I'm a little curious about the "one week to a diet of SMS silage" bit. Was this SMS silage the same as the experimental silages, or a sort of general SMS silage? If it's the latter, could that have introduced any bias?

• In "Collection of spent mushroom substrate (SMS) and silage formulation" section, the authors provide a decent overview of the silage preparation. Table 1 is clear and helpful. One minor thing: perhaps a bit more detail on how the ingredients were mixed to create the micro-silages would be useful. It's probably standard practice, but just to be crystal clear.

• organoleptic characteristics assessment is interesting, and Table 2 is well-defined. My main question here is about the subjectivity of this method. While the scoring system helps, could there be variations in how different people would score the same silage? If so, was any attempt made to minimize this, like having multiple assessors or providing very detailed scoring guidelines?

• chemical analysis section seems pretty standard, which is reassuring. I just wonder if there's a reason why the mycotoxin determination method was "slightly" modified. What were these modifications, and why were they necessary?

• For the in vitro dry matter digestibility (IVDMD) procedure, the authors provide a good level of detail. The go with the flow is good and easy to read.

• In "Collection of spent mushroom substrate (SMS) and silage formulation" section, the units in Table 1 are generally clear. However, for "Ground yellow corn," the unit is "% DM." While technically correct, perhaps just "%" would be more immediately understandable in the context of a formulation percentage. Consistency in unit presentation across the table would be ideal.

• When describing organoleptic characteristics assessment, it might be worth a quick mention of scale used for scoring (e.g., a 1-5 scale, or similar) for readers to better interpret the subsequent results.

• "In vitro digestibility, gas production, and ruminal fermentation kinetics" section, the abbreviation "IVDMD" is used. While it's defined later, might be helpful to spell it out at its first mention in this section for immediate clarity.

Results

• is generally well-organized, presenting the findings clearly and systematically. The use of tables and figures is effective, although Figure 1 could be improved – it's a bit hard to read the labels on the x-axis.

• In "Organoleptic characteristics and pH of the micro-silages" subsection, the authors adequately describe the results from the organoleptic analysis and pH measurements. However, it might be beneficial to include a brief discussion of why these characteristics are important indicators of silage quality. This would provide a bit more context for the reader.

• "Chemical composition of micro-silages" subsection presents the data on nutrient content effectively. One minor point: in some instances, the authors state "no significant differences were observed," which is perfectly fine, but perhaps adding the actual p-values in a few key instances would give the reader a clearer sense of the statistical significance (or lack thereof).

• presentation of the IVDMD, gas production, and VFA results is comprehensive. The authors do a good job of highlighting the key findings and trends.

• Figure 1, as noted in the major comments, the x-axis labels are indeed quite small and difficult to read. Increasing font size or adjusting the layout could improve readability significantly.

• use of "Table" and "Figure" should be consistently capitalized (e.g., "Table 2 shows..." rather than "table 2 shows...").

• When reporting statistical significance (e.g., "P < 0.05"), ensure that the 'P' consistently capitalized and italicized.

Discussion

• is generally well-written and provides a good interpretation of the results. The authors effectively relate their findings to previous research and offer plausible explanations for the observed outcomes.

• However, at some points, the discussion feels a bit repetitive of the Results section. While some reiteration is necessary, the authors should ensure that the Discussion truly expands on the Results by providing more in-depth analysis and interpretation, rather than just summarizing the findings again.

• discussion of the organoleptic properties is quite thorough, which is commendable. The point about the odor of Lentinula edodes being characteristic of the fungi is well-taken.

• authors adequately discuss the implications of their findings for animal feed. but, the discussion could be strengthened by addressing the limitations of the study more explicitly. as example, the authors acknowledge the need for in vivo studies, but they could also discuss any limitations related to the in vitro design itself, or the specific conditions under which the silages were prepared and analyzed.

• When discussing previous research, ensure that citations are placed correctly and consistently. There are a few instances where they appear slightly detached from the text they are supporting.

• language used in the Discussion is generally clear and academic. Just minor point: occasionally, a slightly more concise phrasing could be used to enhance readability. For example, instead of "the fact that...", perhaps just "that..." would suffice.

Conclusion

• is concise and, summarizes the main findings of the study. However, it's a tad too brief. Perhaps a slightly more detailed conclusion that reiterates the significance of the findings and their potential impact would be more impactful.

• is brief, which is acceptable for a conclusion. However, ensure that the key abbreviations used in the paper are consistently used here as well (e.g., SMS).

Figures & Tables

• Ensure all figures and tables are numbered sequentially and are referred to correctly in the main text.

• Check all axes labels in the figures are clear, legible, and include units where appropriate.

• For tables, ensure column headings are clear and that any abbreviations used are defined in the footnotes.

Reviewer #2: Dear Authors,

I have thoroughly examined manuscript PONE-D-25-12392 entitled "Formulation of silages from spent mushroom substrates of Pleurotus ostreatus and Lentinula edodes: organoleptic properties, phenolic content, in vitro digestibility, gas production and ruminal kinetics". The manuscript investigates the potential utilisation of spent mushroom substrate (SMS) as a silage component for ruminant feeding systems. Whilst the topic is certainly meritorious and addresses important questions of agricultural sustainability, I regret to inform you that the manuscript requires substantial revision before it may be considered suitable for publication.

1. The experimental approach suffers from several critical deficiencies. Foremost among these is the absence of appropriate control treatments (lines 94-100). A conventional silage comparator would have provided essential context for evaluating the nutritional and fermentative characteristics of the SMS formulations. The selection of inclusion proportions (70-100% SMS) appears arbitrary and lacks adequate theoretical justification.

2. The organoleptic assessment protocol (lines 105-109) employs a scoring system that has not been validated against established silage quality parameters. Most problematically, the in vitro digestibility assay utilises rumen fluid from goats previously adapted to SMS silage diets (lines 181-183), introducing potential circular bias that may substantially inflate digestibility estimates through prior microbial adaptation.

3. The statistical approach fails to properly account for the factorial structure of the experimental design. The manuscript clearly employs a 4×2 factorial arrangement (four inclusion levels, two fungal species), yet the analysis treats these as eight independent treatments in several instances. No power analysis is presented to justify the selected replication numbers, raising questions about statistical robustness.

4. The authors do not adequately contextualise their findings against established nutritional benchmarks for ruminants. Discussions of phenolic compounds (lines 425-457) and fermentation kinetics (lines 510-547) remain largely descriptive rather than analytical, failing to adequately address the biological significance for rumen function or animal performance. The conclusion overstates the practical applicability based solely on in vitro evidence.

5. The manuscript would benefit from engagement with recent relevant literature, try to synthesize the gap in literatures in particular in lines 48-71. In vitro study may be opportune to be conducted but references below making logic to your current hypothesis especially in tropical country. Particularly, the authors should consider:

1) https://www.mdpi.com/2076-2615/11/4/1053; on "Treatment of Rice Stubble with Pleurotus ostreatus and Urea Improves the Growth Performance in Slow-Growing Goats", which provides valuable in vivo context for fungal treatment applications.

2) Recent work https://www.mdpi.com/2311-5637/8/6/248 on "Optimizing Anthocyanin-Rich Black Cane (Saccharum sinensis Robx.) Silage for Ruminants Using Molasses and Iron Sulphate", which demonstrates a more systematic approach to silage additive optimisation than presented in the current manuscript.

6. Minor recommendations:

1) Include conventional silage control treatments for contextual comparison.

2) Revise the statistical approach to properly account for factorial design.

3) Obtain rumen inoculum from animals not previously exposed to experimental substrates.

4) Strengthen interpretation of findings within the context of practical feeding systems.

5) Consider in vivo palatability trials to address the concerning organoleptic findings, particularly regarding the undesirable odour of L. edodes silages.

Whilst the manuscript presents potentially valuable preliminary data on SMS utilisation in silage production, the aforementioned methodological limitations and interpretative weaknesses substantially undermine its current contribution. I therefore recommend major revision before the manuscript can be considered suitable for publication.

Yours faithfully,

6. PLOS authors have the option to publish the peer review history of their article (what does this mean? ). If published, this will include your full peer review and any attached files.

**Do you want your identity to be public for this peer review?** For information about this choice, including consent withdrawal, please see our Privacy Policy .

Reviewer #1: No

Reviewer #2: No

---

## [Author Response · Author response to Decision Letter 1]

30 Jun 2025

Response to Reviewers

We thank the reviewers and the editor for their thorough evaluation of our manuscript entitled “Formulation of silages from spent mushroom substrates of Pleurotus ostreatus and Lentinula edodes: organoleptic properties, phenolic content, in vitro digestibility, gas production and ruminal kinetics.” We appreciate the constructive comments and helpful suggestions, which have significantly improved the quality and clarity of our work.

Below, we provide a detailed response to each comment. All changes made in the revised manuscript are clearly marked, and specific line numbers are provided where applicable.

Reviewer #1

We thank the reviewer for the time and effort dedicated to evaluating our manuscript and for the constructive comments provided. In the following section, we address the corrections and suggestions made.

Abstract

1. Is generally clear and concise, providing a good overview of the study. However, the phrase "lower production cost has increased significantly" sounds a little awkward. Perhaps rephrasing it to something like "the search for alternative feed sources to lower production costs has increased significantly" would flow better.

Thank you for the suggestion, we rephrase it (line 22-23).

2. It might be helpful to briefly mention the specific percentages of SMS and corn used in the formulations within the abstract to give the reader a quicker grasp of the experimental design.

We added a better explanation of the silage formulations (lines 26-28).

Introduction

1. While does a decent job of setting the stage by discussing the importance of sustainable food production and the potential of spent mushroom substrate (SMS), it feels a bit... rushed, perhaps? It might benefit from a slightly more in-depth exploration of the existing challenges in livestock feed and how SMS can really address them. For instance, elaborating on the environmental and economic benefits a bit more could strengthen the intro. Also, the flow could be smoother; at times, it jumps a bit abruptly between ideas.

R. Thank you for the comment, we included a further explanation of the potential of spent mushroom substrate (SMS)

2. Hypothesis is clearly stated, which is good! However, the justification for focusing on Pleurotus ostreatus and Lentinula edodes specifically could be stronger. Why these two? Are they particularly promising, or were they simply the ones available for the study? Clarifying this would add more context.

R. We added a better explanation for the use of Pleurotus ostreatus and Lentinula edodes

3. When discussing circular economy models, it might be beneficial to briefly explain what these models entail for readers who may not be familiar with the concept in this context. Just a sentence or two could suffice.

R. We explained what these model entail (line 46-48)

4. Use of "(strain Po-IAP)" and "(strain L5)" after mentioning Pleurotus ostreatus and Lentinula edodes is good for clarity. Ensure this consistent throughout the manuscript.

R. We specified the strain after mentioning Pleurotus ostreatus and Lentinula edodes throughout the manuscript.

5. Line 45 I suggest citing 10.1080/19440049.2024.2414954; 10.1038/s41598-025-00675-2 and 10.3389/fvets.2024.1441905 to strength the circular feed sentence

R. Thank you for your suggestions, we improved the introduction and provided more information. We included the suggested citations.

Materials and Methods

1. Description of the animal management and treatments is adequate, and it's good to see the ethics approval mentioned. However, I'm a little curious about the "one week to a diet of SMS silage" bit. Was this SMS silage the same as the experimental silages, or a sort of general SMS silage? If it's the latter, could that have introduced any bias?

R: The silage used for animal adaptation was prepared using the same SMS and following the same methodology used in the experimental silages. Therefore, we believe that no bias was introduced, as the main objective was to promote the adaptation of ruminal microorganisms to the degradation of this ingredient, which is not a regular part of their diet.

2. In "Collection of spent mushroom substrate (SMS) and silage formulation" section, the authors provide a decent overview of the silage preparation. Table 1 is clear and helpful. One minor thing: perhaps a bit more detail on how the ingredients were mixed to create the micro-silages would be useful. It's probably standard practice, but just to be crystal clear.

R: We added a better explanation of the silage formulations (lines 125-132).

3. Organoleptic characteristics assessment is interesting, and Table 2 is well-defined. My main question here is about the subjectivity of this method. While the scoring system helps, could there be variations in how different people would score the same silage? If so, was any attempt made to minimize this, like having multiple assessors or providing very detailed scoring guidelines?

R: We acknowledge that the sensory evaluation method may involve a degree of subjectivity. Therefore, the assessment was carried out by a panel of nine individuals who reached a consensus on each characteristic evaluated for each silage, following an approach like that proposed by Tahuk et al. (2020) and Anas et al. (2024). However, individual evaluations were not documented separately, which is why they are not presented in that form in the manuscript.

Additionally, in vivo testing is planned, during which a more robust sensory analysis will be implemented. Consequently, this phase was considered primarily as an exploratory study.

Anas S, Natsir A, Syahrir. Organoleptic test characteristics of corn stover silage added with several legumes. Hasanuddin J. Anim. Sci. 2024; 6(2):104-113.

Tahuk PK, Bira GF, Taga H, editors. Physical characteristics analysis of complete silage made of sorghum forage, king grass and natural grass. IOP Conference Series: Earth and Environmental Science; 2020: IOP Publishing.

4. Chemical analysis section seems pretty standard, which is reassuring. I just wonder if there's a reason why the mycotoxin determination method was "slightly" modified. What were these modifications, and why were they necessary?

R. Thank you for your observation. To modification were necessary to adapt the method to SMS silage samples. We increased the homogenization time from 1 minute to 5 minutes to ensure complete homogenization of the sample.

5. For the in vitro dry matter digestibility (IVDMD) procedure, the authors provide a good level of detail. The go with the flow is good and easy to read.

R: Thanks for your comment

6. In "Collection of spent mushroom substrate (SMS) and silage formulation" section, the units in Table 1 are generally clear. However, for "Ground yellow corn," the unit is "% DM." While technically correct, perhaps just "%" would be more immediately understandable in the context of a formulation percentage. Consistency in unit presentation across the table would be ideal.

R. The units in table 1 are reported in % for SMS and the ground yellow corn.

7. When describing organoleptic characteristics assessment, it might be worth a quick mention of scale used for scoring (e.g., a 1-5 scale, or similar) for readers to better interpret the subsequent results.

R: Table 2 presents the scale used for the evaluation of organoleptic characteristics. A three-point scale was employed, with values of 0.5, 0.75, and 1, used as a rating system. We considered that, in this case, the use of a five-point scale would have increased the subjectivity of the evaluation, since the main objective was simply to determine whether the characteristics assessed were adequate, fair, or inadequate. This three-point scale is also used by Tahuk et al. (2020), supporting its applicability in this type of assessment.

Tahuk PK, Bira GF, Taga H, editors. Physical characteristics analysis of complete silage made of sorghum forage, king grass and natural grass. IOP Conference Series: Earth and Environmental Science; 2020: IOP Publishing.

8. "In vitro digestibility, gas production, and ruminal fermentation kinetics" section, the abbreviation "IVDMD" is used. While it's defined later, might be helpful to spell it out at its first mention in this section for immediate clarity.

R. We described the meaning of IVDMD at the beginning of the section

Results

1. Is generally well-organized, presenting the findings clearly and systematically. The use of tables and figures is effective, although Figure 1 could be improved – it's a bit hard to read the labels on the x-axis.

R. Thank you for the suggestion, we increased the font size on the x and y axis.

2. In "Organoleptic characteristics and pH of the micro-silages" subsection, the authors adequately describe the results from the organoleptic analysis and pH measurements. However, it might be beneficial to include a brief discussion of why these characteristics are important indicators of silage quality. This would provide a bit more context for the reader.

R: A brief explanation of the relevance of these characteristics was added in line 282-254. However, it is important to note that each of these aspects is addressed in detail in the discussion section; therefore, they were not elaborated upon again in the results to avoid redundancy in the text.

3. "Chemical composition of micro-silages" subsection presents the data on nutrient content effectively. One minor point: in some instances, the authors state "no significant differences were observed," which is perfectly fine, but perhaps adding the actual p-values in a few key instances would give the reader a clearer sense of the statistical significance (or lack thereof).

R. Thank you for the suggestion, we included the p value of all the results.

4. presentation of the IVDMD, gas production, and VFA results is comprehensive. The authors do a good job of highlighting the key findings and trends.

R: Thanks for your comment

5. Figure 1, as noted in the major comments, the x-axis labels are indeed quite small and difficult to read. Increasing font size or adjusting the layout could improve readability significantly.

R. Thank you for the suggestion, we increased the font size on the x and y axis in the figures

6. use of "Table" and "Figure" should be consistently capitalized (e.g., "Table 2 shows..." rather than "table 2 shows...").

R. All Tables and Figures are capitalized.

7. When reporting statistical significance (e.g., "P < 0.05"), ensure that the 'P' consistently capitalized and italicized.

Discussion

1. is generally well-written and provides a good interpretation of the results. The authors effectively relate their findings to previous research and offer plausible explanations for the observed outcomes.

2. However, at some points, the discussion feels a bit repetitive of the Results section. While some reiteration is necessary, the authors should ensure that the Discussion truly expands on the Results by providing more in-depth analysis and interpretation, rather than just summarizing the findings again.

3. discussion of the organoleptic properties is quite thorough, which is commendable. The point about the odor of Lentinula edodes being characteristic of the fungi is well-taken.

4. authors adequately discuss the implications of their findings for animal feed. but, the discussion could be strengthened by addressing the limitations of the study more explicitly. as example, the authors acknowledge the need for in vivo studies, but they could also discuss any limitations related to the in vitro design itself, or the specific conditions under which the silages were prepared and analyzed.

5. When discussing previous research, ensure that citations are placed correctly and consistently. There are a few instances where they appear slightly detached from the text they are supporting.

6. language used in the Discussion is generally clear and academic. Just minor point: occasionally, a slightly more concise phrasing could be used to enhance readability. For example, instead of "the fact that...", perhaps just "that..." would suffice.

We thank the reviewer for their constructive feedback on the Discussion section. In response:

• We reduced redundancy by avoiding direct repetition of Results and instead emphasized interpretive and comparative analysis.

• The discussion was reorganized to enhance thematic coherence and biological

• We reviewed and corrected all citation placements to ensure consistency and clarity.

• Finally, we revised phrasing throughout the section to improve conciseness and readability, following the reviewer’s suggestions.

Conclusion

1. is concise and, summarizes the main findings of the study. However, it's a tad too brief. Perhaps a slightly more detailed conclusion that reiterates the significance of the findings and their potential impact would be more impactful.

R. Thank you for your observation. We have revised the conclusion to include a more detailed summary of the key findings and to emphasize the potential significance of SMS silage as a sustainable feed resource.

2. is brief, which is acceptable for a conclusion. However, ensure that the key abbreviations used in the paper are consistently used here as well (e.g., SMS).

R. Thank you for pointing this out. We have ensured that all key abbreviations used throughout the manuscript, including SMS, are consistently applied in the conclusion.

Figures & Tables

1. Ensure all figures and tables are numbered sequentially and are referred to correctly in the main text.

R. We appreciate this comment. All figure axes have been checked to confirm that labels are clear and legible, and units have been included where applicable.

2. Check all axes labels in the figures are clear, legible, and include units where appropriate

3. R. Thank you for the suggestion. We have reviewed the entire manuscript to ensure that all figures and tables are numbered sequentially and are correctly cited in the main text.

4. For tables, ensure column headings are clear and that any abbreviations used are defined in the footnotes.

R. Thank you for your observation. We have revised all tables to ensure that column headings are unambiguous, and all abbreviations used have been defined in the corresponding table footnotes. 

Reviewer #2: Dear Authors,

I have thoroughly examined manuscript PONE-D-25-12392 entitled "Formulation of silages from spent mushroom substrates of Pleurotus ostreatus and Lentinula edodes: organoleptic properties, phenolic content, in vitro digestibility, gas production and ruminal kinetics". The manuscript investigates the potential utilization of spent mushroom substrate (SMS) as a silage component for ruminant feeding systems. Whilst the topic is certainly meritorious and addresses important questions of agricultural sustainability, I regret to inform you that the manuscript requires substantial revision before it may be considered suitable for publication.

1. The experimental approach suffers from several critical deficiencies. Foremost among these is the absence of appropriate control treatments (lines 94-100). A conventional silage comparator would have provided essential context for evaluating the nutritional and fermentative characteristics of the SMS formulations. The selection of inclusion proportions (70-100% SMS) appears arbitrary and lacks adequate theoretical justification.

R: Thank you for this insightful comment. Maize silage is widely recognized as the most used silage worldwide due to its high yield potential and favorable fermentation characteristics. However, it also presents certain limitations, as noted in lines 404-406, which have prompted the search for alternative silage materials. In this context, the use of non-maize silages has gained increasing relevance, leading to numerous studies in which maize silage is no longer necessarily used as a conventional comparator (Barbosa et al., 2025; Jalil et al., 2024; Yi et al., 2023; Wu et al., 2020). This shift is partly because the characteristics of maize silage have been extensively documented in scientific literature, making them a suitable reference rather than a direct point of comparison. Moreover, the primary objective of silages produced from alternative feedstocks is not to replicate the fermenta

---

## [Decision Letter · Decision Letter 1]

14 Jul 2025

PONE-D-25-12392R1Formulation of silages from spent mushroom substrates of Pleurotus ostreatus and Lentinula edodes: organoleptic properties, phenolic content, in vitro digestibility, gas production and ruminal kineticsPLOS ONE

Dear Dr. Márquez-Mota,

Thank you for submitting your manuscript to PLOS ONE. After careful consideration, we feel that it has merit but does not fully meet PLOS ONE’s publication criteria as it currently stands. Therefore, we invite you to submit a revised version of the manuscript that addresses the points raised during the review process.

We look forward to receiving your revised manuscript.

Kind regards,

Agung Irawan

Academic Editor

PLOS ONE

Journal Requirements:

Additional Editor Comments:

Dear authors, 

Please consider the comments from reviewer 1. In addition, why the chemical composition of the silage and in vitro gas production data were analyzed differently and separately? By design, they appear to be similar and should be treated/ analyzed at the same way. 

Please also check thoroughly for language/ grammar accuracy and typos in your manuscript when resubmitting your revision. 

Reviewers' comments:

Reviewer's Responses to Questions

**Comments to the Author**

1. If the authors have adequately addressed your comments raised in a previous round of review and you feel that this manuscript is now acceptable for publication, you may indicate that here to bypass the “Comments to the Author” section, enter your conflict of interest statement in the “Confidential to Editor” section, and submit your "Accept" recommendation.

Reviewer #1: All comments have been addressed

Reviewer #2: All comments have been addressed

2. Is the manuscript technically sound, and do the data support the conclusions?

Reviewer #1: Yes

Reviewer #2: Partly

3. Has the statistical analysis been performed appropriately and rigorously? 

Reviewer #1: Yes

Reviewer #2: Yes

4. Have the authors made all data underlying the findings in their manuscript fully available?

Reviewer #1: Yes

Reviewer #2: Yes

5. Is the manuscript presented in an intelligible fashion and written in standard English?

Reviewer #1: (No Response)

Reviewer #2: Yes

6. Review Comments to the Author

Reviewer #1: The authors addressed all the points, I have some minor comments left:

Materials and Methods

Include conventional silage control treatments for contextual comparison. This could help provide a better baseline for how your SMS formulations perform.

You should revise the statistical approach to properly account for the factorial design. It seems like in some parts, the analysis might be treating things as separate when they are part of a larger experimental setup.

When getting rumen inoculum, it would be good to obtain it from animals that haven't been exposed to the experimental substrates before. This could help avoid any potential biases in the results.

Discussion

Try to strengthen the interpretation of your findings within the context of practical feeding systems. Making it clearer how this research applies to real-world scenarios would be beneficial.

You might want to consider in vivo palatability trials to address the concerning organoleptic findings, especially regarding the undesirable odor of

L. edodes silages. This could provide valuable insights into how animals actually perceive the feed.

Reviewer #2: Dear authors,

I notice your big effort in this revision, making the current manuscript appropriately acceptance.

7. PLOS authors have the option to publish the peer review history of their article (what does this mean? ). If published, this will include your full peer review and any attached files.

**Do you want your identity to be public for this peer review?** For information about this choice, including consent withdrawal, please see our Privacy Policy .

Reviewer #1: No

Reviewer #2: No

---

## [Author Response · Author response to Decision Letter 2]

13 Aug 2025

PLOS ONE editor

Thank you for your feedback and revision of the manuscript titled "Formulation of silages from spent mushroom substrates of Pleurotus ostreatus and Lentinula edodes: organoleptic properties, phenolic content, in vitro digestibility, gas production and ruminal kinetics" (PONE-D-25-12392).

As you indicated, the data on in vitro gas production and chemical composition come from a single experiment. However, because of their disparate data structures and research goals, the data could not be examined using the same statistical model:

• Since each analysis aimed to look at the nutritional profile of each silage treatment on its own, we used one-way ANOVA to study the chemical composition and phenolic content. Since treatment was the primary factor and we were interested in the properties of the formulated silages, we did not consider the impact of the interaction between the levels of inclusion or the type of SMS.

• On the other hand, data on ruminal kinetics, digestibility, and in vitro gas production were gathered over time and necessitated a factorial approach (4 levels of SMS inclusion × 2 mushroom species). To consider the main effects as well as the interaction effects of species and inclusion levels on dynamic fermentative outcomes, these variables were examined using a 2-way ANOVA under a 4 × 2 factorial design.

Response to Reviewers

We thank the reviewer and the editor for their thorough evaluation of our manuscript entitled “Formulation of silages from spent mushroom substrates of Pleurotus ostreatus and Lentinula edodes: organoleptic properties, phenolic content, in vitro digestibility, gas production and ruminal kinetics.” We appreciate the constructive comments and helpful suggestions, which have significantly improved the quality and clarity of our work.

Below, we provide a detailed response to each comment. All changes made in the revised manuscript are clearly marked, and specific line numbers are provided where applicable.

Reviewer #1

We thank the reviewer for the time and effort dedicated to evaluating our manuscript and for the constructive comments provided. In the following section, we address the corrections and suggestions made.

Materials and Methods

1. Include conventional silage control treatments for contextual comparison. This could help provide a better baseline for how your SMS formulations perform.

R: Thank you for your suggestion. Maize silage is widely recognized as the most used silage worldwide due to its high yield potential and favorable fermentation characteristics. However, it also presents certain limitations, as noted in lines 404-406, which have prompted the search for alternative silage materials. In this context, the use of non-maize silages has gained increasing relevance, leading to numerous studies in which maize silage is no longer necessarily used as a conventional comparator (Barbosa et al., 2025; Jalil et al., 2024; Yi et al., 2023; Wu et al., 2020). This shift is partly because the characteristics of maize silage have been extensively documented in scientific literature, making them a suitable reference rather than a direct point of comparison. Moreover, the primary objective of silage produced from alternative feedstocks is not to replicate the fermentation profile of maize silage, but rather to achieve a product with adequate fermentation quality, that preserves its nutritional attributes and is suitable for animal feeding.

Reference added:

Barbosa CR, Santos MM, Meirelles PR, et al. (2025). Elephant grass silage with pelleted citrus pulp: Chemical composition, disgestibility, and feedlot costs. Tropical Animal Science Journal, 48(2):148-155.

Jalili M, Mohammadzadeh H, Hosseinkhani A, Taghizadeh A. (2024). Effect of bacterial, enzymatic and molasses additive on fermentation and nutrient composition of Lime (Citrus aurantifolia) Pulp Silage. Animal Science Research, ():-. doi: 10.22034/as.2023.50768.1653

Yi Q, Wang P, Tang H, et al. (2023). Fermentation quality, in vitro digestibility, and aerobic stability of ensiling spent mushroom substrate with microbial additives. Animals, 13(5),920. https://doi.org/10.3390/ani13050920

Wu P, Li L, Jiang J, et al. (2020). Effects of fermentative and non-fermentative additives on silage quality and anaerobic digestion performance of Pennisetum purpureum. Bioresource Technology, 297, 122425.

2. You should revise the statistical approach to properly account for the factorial design. It seems like in some parts, the analysis might be treating things as separate when they are part of a larger experimental setup.

R. As you indicated, the data on in vitro gas production and chemical composition come from a single experiment. However, because of their disparate data structures and research goals, the data could not be examined using the same statistical model:

• Since each analysis sought to characterize the nutritional profile of each silage treatment separately and independently, one-way ANOVA was used to analyze the chemical composition and phenolic content. Since treatment was the primary factor and we were interested in the properties of the formulated silages, we did not consider the impact of the interaction between the levels of inclusion or the type of SMS.

• On the other hand, data on ruminal kinetics, digestibility, and in vitro gas production were gathered over time and necessitated a factorial approach (4 levels of SMS inclusion × 2 mushroom species). To take into consideration the main effects as well as the interaction effects of species and inclusion levels on dynamic fermentative outcomes, these variables were examined using a 2-way ANOVA under a 4 × 2 factorial design.

3. When getting rumen inoculum, it would be good to obtain it from animals that haven't been exposed to the experimental substrates before. This could help avoid any potential biases in the results.

R. Thank you for the comment. We recognize the potential concern regarding microbial adaptation bias. However, the use of rumen fluid from animals adapted to the test feed is consistent with standard protocols in in vitro digestibility assays, as established by Goering and Van Soest (1970) and Tilley and Terry (1963). This practice ensures that the microbial population is metabolically active and relevant to the substrate under study, thereby improving the biological relevance and repeatability of the assay.

Importantly, the goal of our assay was to compare the relative digestibility of different SMS-based formulations, not to generalize absolute digestibility values to all feeding conditions. The use of adapted rumen fluid helps standardize conditions across treatments.

References Cited:

• Tilley JMA, Terry RA. (1963). A two‐stage technique for the in vitro digestion of forage crops. Journal of the British Grassland Society, 18(2), 104–111.

• Goering HK, Van Soest PJ. (1970). Forage Fiber Analyses (Apparatus, Reagents, Procedures, and Some Applications). USDA Agricultural Handbook No. 379.

Discussion

4. Try to strengthen the interpretation of your findings within the context of practical feeding systems. Making it clearer how this research applies to real-world scenarios would be beneficial.

R. Thank you for your suggestion to strengthen the interpretation of our findings within practical feeding systems. We included a paragraph in the discussion section to ensure the applicability of our research to real-world animal production contexts (Line 681-688).

5. You might want to consider in vivo palatability trials to address the concerning organoleptic findings, especially regarding the undesirable odor of L. edodes silages. This could provide valuable insights into how animals actually perceive the feed.

R. We appreciate this comment. The present study is an in vitro study the provided a first approach of the possible use of silage formulated with SMS from L. edodes and P. ostreatus. The organoleptic evaluation indicated an undesirable and potentially unpalatable odor in the L. edodes silages, which may negatively influence animal acceptance

We addressed this limitation and included in the discussion section the necessity of in vivo palatability trials to evaluate the animal response and feed intake behavior (Line 453-455).

---

## [Editor Report · Decision Letter 2]

17 Aug 2025

Formulation of silages from spent mushroom substrates of Pleurotus ostreatus and Lentinula edodes: organoleptic properties, phenolic content, in vitro digestibility, gas production and ruminal kinetics

PONE-D-25-12392R2

Dear Dr. Márquez-Mota,

We’re pleased to inform you that your manuscript has been judged scientifically suitable for publication and will be formally accepted for publication once it meets all outstanding technical requirements.

Kind regards,

Agung Irawan

Academic Editor

PLOS ONE
---

## [Editor Report · Acceptance letter]

PONE-D-25-12392R2

PLOS ONE

Dear Dr. Márquez-Mota,

I'm pleased to inform you that your manuscript has been deemed suitable for publication in PLOS ONE. Congratulations! Your manuscript is now being handed over to our production team.

Kind regards,

on behalf of

Dr. Agung Irawan

Academic Editor

PLOS ONE